# Cross-cultural structures of personal name systems reflect general communicative principles

Michael Ramscar [1] ✉, Sihan Chen[2], Richard Futrell [3] & Kyle Mahowald[4]

The structure of personal names appears to differ widely across cultures. Using census records and historical datasets, we present an information-theoretic analysis of name systems that shows how the scope of this variation is more constrained than it might appear. We identify two constraints name systems must satisfy: encoding large numbers of identities, and ensuring these encodings are usable. We show that, historically, the world's languages satisfied these constraints using structurally similar, near-optimal codes. They did so by combining sets of name-specific words with existing vocabulary items, allowing unlimited numbers of identifiers to be created while keeping vocabulary sizes stable. Today, many natural name systems have been transformed into official codes based on hereditary patronyms. We show how, globally, these changes differentially altered the information structure of codes, leading to cross-cultural differences in the way names function as individuators that can have tangible effects in domains like scientific publishing.

Personal names (which may comprise a word or set of words to form an identifier) provide a means for constructing identities and signaling information about individuals. For this latter function to be achieved, name systems must provide encodings that can allow individuals to be identified in communication, while also being organized to enable these encodings to be shared and used. An obvious strategy for achieving the first requirement would be to use a unique name word for each individual. However, although this approach might work for the 82 principal characters in "Game of Thrones" (who almost all have unique first names), it would fail the second requirement in any large-scale modern society, where it would require millions of pronounceable yet not already used words to be generated. Given the limited coding resources of natural languages, many names would end up being both very long and highly confusable, making the resulting system impossible to learn or use. Perhaps unsurprisingly, instead of employing unique name words, cultures satisfy the lexical demands of naming by creating combinatoric identifiers[1–4]. These combine small sets of lexical items that are often name specific with preexisting vocabulary elements to encode individual identities as phrases, as in the names of the badminton player Simon Archer and the basketball player Yao Ming 姚明 (Simon and Yao are names, whereas Archer is an English word for someone who shoots arrows with a bow, and Ming is a Chinese word meaning brightness).

We present an information-theoretic analysis of personal names that reveals historical commonalities in the systems of combinatoric identifiers used in different cultures, following the approach in ref. 1. We use this analysis to examine the different ways in which the legalization of names—which occurs as societies become more organized and bureaucratic[5] – has changed these systems, and show how legalization can impede the communicative efficiency of name systems by imposing top-down structures on what had been organically evolved systems. Finally, taking international scientific publishing as an example, we show how the adoption of arbitrary standards that are blind to these differences can differentially impair the way the names of authors from different cultures function as individuators.

Since our focus will be on the coding function and information structure of name systems, we describe the parts of name-phrases that are spoken or written first as 'prefix-names', and the parts that follow them as 'bynames' (see Fig. 1). Prefix-names can be specific lemmas (proprial lemmas[2]) or other vocabulary items marked by affixes[6]. The

[1]Tübingen University, Baden-Württemberg, Germany. [2]Massachusetts Institute of Technology, Cambridge, MA, USA. [3]University of California, Irvine, CA, USA. [4]The University of Texas at Austin, Austin, TX, USA. ✉e-mail: michael.ramscar@uni-tuebingen.de

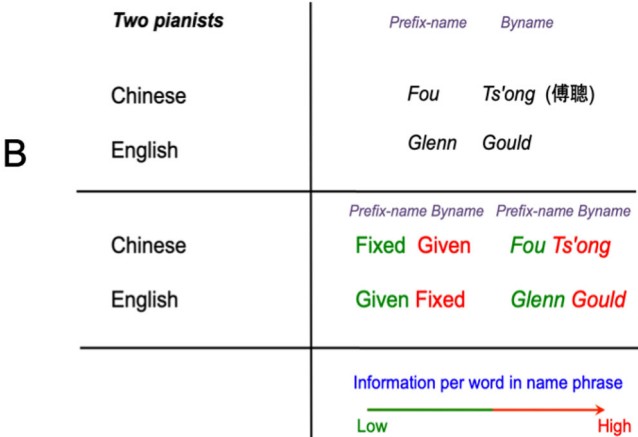

**Fig. 1 | Western names and East Asian names, despite differences in naming traditions, share a similar information structure. A** Two forms of the same name, movie director Woo Yu-Sen / John Woo. In Mandarin, the (inherited) prefix-name Woo comes first in speech. However, in its conventional Westernized form, Woo becomes the byname. This convention aligns the given and fixed (inherited) parts of EACS and Western names (because Western prefix-names are given and bynames inherited, whereas EACS bynames are given and prefix-names inherited), however it misaligns their information structures. In both Western and EACS name-phrases, prefix-names are systematically less informative than bynames, thus in (**B**): Chinese pianist Fou Ts'ong's inherited name is his prefix-name Fou, while Canadian pianist Glenn Gould's inherited name is his byname Gould; Fou's given name is his byname Ts'ong, whereas Gould's given name is his prefix-name Glenn.

semantics of lemma prefix-names are typically opaque, such as John or Mary in English, or *Liú* 刘 in Chinese[2], or else they are irrelevant in use, as in the English names Victor, Rose or Destiny. When affixing is used, for example, in Ngəmbà, names are made by marking preexisting vocabulary items with prefixes, such as Mà for females and Tà for males[7]. Again, marking existing words as names changes their semantics, and often renders them irrelevant in use.

When it comes to bynames, across cultures name systems have largely either re-employed prefix-names for this purpose (e.g., Rhys ap Llywelyn – where ap means 'son of' – is a fairly typical patronym in the Welsh naming system[8]; see also Johnson in English, Jónsdóttir in Icelandic, etc.) or else re-deployed existing vocabulary items. For example, *Hóng* 红, Goch, Roth, and Červený, which have often been used as bynames in Mandarin, Welsh, German, and Czech, respectively, are simply forms or derivatives of the word for red. From a communicative perspective, this is notable because it enabled these systems to generate enormous numbers of individual identifiers while simultaneously avoiding the need to add huge numbers of extra words to linguistic inventories.

The relative benefits of redeploying old words versus adding new words in this way can be operationalized and measured using information theory[9], which provides a means for formalizing notions such as information and efficiency in communication, and has been

fruitfully applied to a variety of questions in linguistics[10–14] by treating languages as information-theoretic codes. Formally, information (or entropy) is defined in terms of the uncertainty associated with an event or distribution of events. Efficient codes minimize the complexity of codewords while maximizing their informativity about underlying messages[9]. Prefix-names seem to offer this kind of efficiency: If 33% of people are named John, and 1% Gerald, Gerald will be more informative than John (Gerald better discriminates individuals than John). Simultaneously, not only will John be easier to recognize and process[15], but the large number of Johns will reduce the total number of names, making the system as a whole easier to learn and use. Intuitively, a system where frequent prefix-names like John are distinguished by further optional codewords conditioned on them (i.e., John Adams, John Hancock...) may seem simpler and more efficient than a system in which everyone has a unique name. Information theory provides tools for assessing whether this is the case or not (and assessing the impact of changing the relative numbers of Johns and Geralds in systems).

Historically, across cultures, prefix-names served as the primary historical means for referring to individuals[5,16]. However, these names were not always fixed in the modern sense. Rather, the name by which an individual was known depended on the context in which it was used. This tendency is still evident in indigenous Australian prefix-names, which can change as people age or undergo major life events[17], or Yupik and Inuit prefix-names, which can change throughout a person's life (and have even been documented as changing after a serious illness[18]). It was also evident in the vernacular names of pre-modern Europe, which were similarly context-dependent[2]: the same "John" might have been "John (the) Smith" to distinguish him from "John (the) Baker" but "John Short" in other contexts[5]. In use, it even remains true for modern U.S. names: e.g., the 44th President of the U.S. is "Barack Hussein Obama" on his birth certificate, "Barry" (a childhood nickname) to intimates, "Barack" to others, "bo" on Twitter, etc.

All of which raises a question: why have the context-dependent vernacular name systems that characterize most of human history now been replaced by official, fixed name systems? The answer seems to lie in the way that the context-dependence of vernacular names conflicted with the needs of bureaucracies and governments which, as they developed, required that their citizenries be ever more "legible"[5] (i.e., that their activities be amenable to top-down observation and management). Fixed – context-free – official names offered bureaucracies an objective, tractable technology for managing citizens and their property[5], making tasks such as taxation, conscription, etc., far more manageable. Historically, this conflict is most clearly visible in the laws that were used to fix the naming practices of particular populations, such as those that led to the enforcement of patronymic-bynames on Native Americans[19], Jewish people in Europe[5], and on the Inuit population in Canada (including attempts to assign each individual a unique number[20]). Meanwhile, the resolution of this conflict in favor of the needs of governments can be seen in the fact that today most states have legalized their citizens' names: across the globe, the vernacular naming practices that may be as old as languages themselves have now been replaced with official, fixed name codes. Palsson[20] summarizes it starkly: "Clashes [...] between different traditions and practices of naming, especially in the context of slavery and empire, illuminate with clarity the relevance of names as technologies of exclusion and belonging".

Critically, the legal imposition of fixed names (as hereditary patronyms) was implemented differently in the East Asian Cultural Sphere (EACS) as compared to the West. In particular, whereas laws in the EACS made prefix-names hereditary (and thus fixed), Western laws transformed bynames into hereditary, fixed patronyms (Fig. 1). These differences provide a natural way of exploring whether name systems serve to communicate identities in the way we propose, because our information-theoretic analysis provides clear predictions about the way we should expect vernacular name systems to have been

structured across cultures historically, and how they should subsequently have developed over time, given the changes that have been imposed.

Notably, from this perspective, it has been shown that the sets of prefix-names employed in China were historically quite small (in China, lǎobǎixíng 老百姓 – "old hundred names" – is the colloquial term for the masses, such that historically 50% of the population shared just six of them[5,21]) with their overall distribution being exponential[22]. Not only does this provide an efficient way of organizing variable-length codewords[23], but it has also been shown that Korean prefix-name distributions had (and have) similar properties[24,25]. It is thus notable that in every 50 year period between 1550–1800, records reveal that roughly 50% of the UK population shared 3 female prefix-names (Elizabeth, Anne and Mary) and 3 male prefix-names (John, William and Thomas)[26], and that the stocks of prefix-names in communities also typically approximated 100[27]. Other historical prefix-name distributions across Europe were similar[26].

In China, people were assigned patronymic fixed hereditary prefix-names after population registration efforts in the Qin dynasty[28], and similar systems later developed in other EACS states. In contrast, when European nation states issued laws that fixed names from the 18th century onward, they turned bynames (already associated with property inheritance) into fixed hereditary patronyms (hereditary bynames)[5]. Importantly, because bynames rather than prefix-names were fixed in the West, it followed that if populations were to grow – as they did in response to industrialization – then the total number of unique identities encodable in European name-systems could only be increased by adding more prefix-names, the information structure of which appears to have been generally stable across Western cultures up to this point[25,26].

Here, we empirically test several hypotheses related to the communicative efficiency of names. In our first study, we collect and analyze the properties of historical name data from birth records, census records, and other historical sources. Our first key hypothesis is that, given that Western cultures largely retained vernacular name systems until the 19th century, we should expect pre-modern prefix-names distributions in these cultures to reflect this common system and show relatively low entropy on prefix-names. First, we establish that vernacular name systems across cultures show a similar low-information profile. Moreover, in countries where prefix-names were hereditary fixed patronyms—i.e., EACS countries where laws served to fix a small relatively invariant information system – we should expect distributions of prefix-names to largely resemble vernacular name systems, regardless of any subsequent population increases. Thus, empirically, we should expect modern distributions of EACS prefix-names to resemble those of pre-modern Western cultures. On the other hand, we expect that, once hereditary bynames were implemented in the West, as populations grew, Western prefix-name distributions would shift to communicate more information in proportion to this growth. In Study 1, we confirm these hypotheses using historical data.

In Study 2, to provide a more detailed test, we examine a dataset of church records from Finland (in the period roughly 1700 to 1900, a period during which the population grew from less than 500,000 people to over 2.5 million). These provide a particularly useful means for testing our proposal, first because hereditary fixed bynames were adopted during this time, and second because their adoption was asymmetrical across the country, occurring first in eastern Finland and then increasingly in the west of Finland.

Accordingly, given that this system enables us to observe the introduction and development of the use of hereditary fixed bynames in a name system, based on our first hypothesis, we predicted that we would observe a corresponding increase in the entropy of prefix-names across this period. Further, we predicted that if the increase in information in prefix-names is a result of the transformation of flexible bynames into hereditary fixed patronyms, prefix-name information entropy should increase as a function of the degree to which fixed

hereditary bynames are used. To foreshadow our findings, we observe the predicted correlation with the entropy of Finnish prefix-names both over space (higher entropy first names in the east) and time (higher entropy first names later in time, as hereditary fixed bynames become more prevalent).

Taken together, these results enable us to derive another main hypothesis – which we test in Study 3. We have argued that naming systems have evolved to yield name phrases that have low-information prefix-names and high-information bynames, and shown how different cultures have made different parts of these phrases into hereditary fixed patronyms. Accordingly, it follows that forcing names from different cultures into alignment according to their hereditary patronymics as opposed to their information structure (i.e., making names from EACS countries adopt the Western convention of putting the hereditary patronym at the end of a name phrase) could undermine the communicative function of some name-phrases as compared to others.

We show that this is the case in academic publishing in the Anglophone world by considering a sample of names from academic honor societies from 6 countries (China, Korea, Finland, France, Russia, and the U.S.). Under the Western convention, publications are often cited in papers by the hereditary patronym of the first author (where this is a high-information byname, such as Einstein et al.), whereas in China and Korea, publications are often cited by the full name of the first author (e.g., Li Zhaoping et al.). As predicted, we show that forcing names from China and Korea into the Anglophone model, in which only the hereditary patronym is given in full (which in both cases is a low-information prefix-name) and in which bynames are initialized (e.g., X. Li) induces unnecessary name ambiguity that can be resolved if other conventions are employed.

## Results

### Study 1: Name distributions across culture and time

Our first hypothesis predicted that the distribution of prefix-names in pre-modern Western populations (i.e., before bynames were turned into fixed patronyms) would have been similar, communicating relatively low levels of information (operationalized as Shannon entropy[9]); that they will resemble modern EACS prefix-names (which are themselves hereditary patronyms); and that modern Western prefix-names will have communicated increasingly higher levels of information as populations grew.

To test this, we examined the prefix-name distributions of birth-records in four pre-modern Scottish parishes: Earlston, Govan, Dingwall, Beith, between 1700–1800[27], analyzed individually and collectively. We also examined the birth records from Finland from the same period, along with marriage registers in the counties of Northumberland and Durham in Northern England (again covering the same period[29]). Our EACS prefix-name distributions were taken from 2015 South Korean census data[30], 2018 Taiwanese census data[31], and Vietnamese-American population data extracted from the 2010 U.S. census[32]. It is worth noting that although prefix-names are patronyms in Taiwan, Korea, and Vietnam, the languages spoken in the three countries belong to different linguistic families: many of the major languages spoken in China and Taiwan are part of the Sino-Tibetan language family,[33] modern Korean is a language isolate[34], and Vietnamese is an Austroasiatic language, albeit with significant vocabulary borrowings from China[35].

For our modern Western analyses, we used post-1900 data taken from a larger genealogical set of Finnish name records[36], covering the period after hereditary patronyms had become common in Finland[37], and data from the United States Social Security records between 1910 and 2010[38], focusing on a representative large state, California, and a small state, Delaware. Table 1 lists the number of names and population in each dataset (see ref. 1 for a smaller-scale analysis comparing just the US and Korea). No data were excluded from the analyses.

**Table 1 | Summary of several unique names and the entropy estimates of the prefix-name distribution in each dataset**

| Locale | Name count | Cut-offs | Sample | LB | UB |
|---|---|---|---|---|---|
| California | 17638 | 5 | 27517342 | 10.16 | 14.40 |
| Delaware | 1534 | 5 | 597042 | 8.57 | 12.72 |
| All US, 1910–2010 | 28638 | 5 | 286605557 | 9.80 | 11.22 |
| Finland Post-1900 | 1066 | 1 | 75814 | 7.20 | 7.20 |
| Taiwan | 500 | 135 | 23543563 | 5.68 | 5.71 |
| Finland Pre-1800 | 1833 | 1 | 1748523 | 5.43 | 5.43 |
| Korea | 533 | 5 | 49705663 | 4.82 | 5.41 |
| Northern England | 225 | 1 | 25797 | 4.93 | 4.93 |
| Dingwall | 81 | 1 | 1685 | 4.87 | 4.87 |
| Earlston | 79 | 1 | 3121 | 4.85 | 4.85 |
| Govan | 121 | 1 | 12086 | 4.70 | 4.70 |
| Scottish | 361 | 1 | 23792 | 4.65 | 4.65 |
| Vietnamese-American | 62 | 100 | 1527654 | 4.35 | 4.59 |
| Beith | 80 | 1 | 6900 | 4.35 | 4.35 |

The inclusion threshold, namely the least frequent prefix-name included, along with the total number of people sampled, is included. The lower bound entropy (LB in the table) reflects the observed sample (in bits) while the upper bound (UB in the table) reflects an estimate assuming that all unobserved names are unique (i.e., it assumes the maximum entropy for the unobserved portion of the distribution).

The information communicated by each prefix-name distribution was calculated as follows: if there are $N$ prefix-names in a given distribution, each with a frequency of $p_i$, the entropy of the distribution is:

$$H = -\sum_{i=1}^{N} p_i \log_2 p_i. \tag{1}$$

Note that although there is a large amount of data on personal names and their distributions in existence, studying it is complicated by several related factors: name data tend to be the preserve of state governments, and responsible governments require that publicly released data relating to individuals be anonymized[39]. This last point necessarily conflicts with the sui generis function of names – and especially fixed, legally defined names – which make the identification of many individuals from full name data trivial. Accordingly, most of our data do not contain the prefix-name of every individual and contains a cut-off number, reflecting these concerns (see also Study 1 in the Methods Section). If the number of people sharing a prefix-name is lower than the cut-off, the prefix-name is not part of the data. Therefore, the entropy calculated from our data is only a lower bound of the true entropy of the population. To estimate an upper bound, we assume an extreme (if unlikely) scenario, namely that everyone not included in the database has a different prefix-name. For example, the 2015 Korean census data do not contain the prefix-names of approximately 1.1 million people. To estimate an upper bound of Korean prefix-name entropy, we assume these 1.1 million people have 1.1 million different prefix-names and calculate the entropy of the population under this assumption.

Figure 2 plots the 100 most common prefix-names (dividing the frequency of each name by the frequency of the most common name) from each distribution. As predicted (Table 1), the different prefix-name distributions in our pre-modern Western samples communicated relatively small – and similar – amounts of information: Earlston, Govan, Dingwall, Beith have entropies of 4.85, 4.70, 4.87, and 4.35 bits, respectively (4.65 bits when aggregated together); the Northern English

sample had an entropy of 4.93 bits; and the Finnish prefix-name distribution taken from the same period had an entropy of 5.43 bits.

Also as predicted, the entropies of our Vietnamese American and Korean samples (where 92% of the population share the 50 most common prefix-names) were in the same range (4.35 and 4.82 bits, respectively), as was the entropy of our sample of Taiwanese prefix-names (5.68 bits).

Critically, the US name distributions have by far the highest entropies (10.16 bits in California and 8.57 bits in Delaware), and the prefix-name entropy in our Finnish sample rises to 7.20 bits after 1800, approaching the range observed in our US data (this change is analyzed in detail below). It should be emphasized that higher entropies in these results do not simply reflect larger population sizes: the Korean sample is far bigger than the Delaware sample, but even the upper bound of the Korean name entropy is much lower than the lower bound of the name entropy for Delaware. In addition, results from a study on Korean family records[25] indicated that, despite a large growth in population between 1510 and 1990, and the inclusion of many prefix-names in this period, prefix-name entropy remains largely unchanged. Similarly, the entropy of our aggregated Scottish sample is lower than the individual entropies of 3 of the 4 parishes it comprises.

## Study 2: Relationship between fixing patronyms and prefix-name distributions

Since industrialization, the world's population has grown considerably[40]. The population of 19th-century Britain grew rapidly[41] and as it did, the distribution of prefix-names changed: in 1800, >50% of children were given the most frequent 6 prefix-names; by 1900, it was <25% (Fig. 2). This pattern is consistent with prefix-name information increasing to offset the development of hereditary fixed bynames. However, to make a more detailed assessment of these changes we examined a set of Finnish birth records covering the period 1600-1917 (from the same dataset examined in Study 1[36]). Because of the late implementation of name laws in Finland (1929), and the fact that hereditary fixed bynames became dominant in eastern Finland earlier than western Finland[37], these data offer a unique opportunity to explore the association between information profiles in flexible byname systems as opposed to in hereditary patronymic-byname systems, for a system in transition. We divided the records, which comprise a baby's prefix-name, parish of birth and, where available, a paternal byname (e.g., relating to the farm or village of the father) into 5 different year bins: pre-1730, 1730–1780, 1780–1830, 1830–1880, and post-1880. Due to the uneven number of records in each birth year bin (Table 2), from each birth year bin and parish we randomly sampled 50 births, from 473 parishes.

Taking the rates of occurrence of paternal bynames in records as a proxy for the use of hereditary fixed bynames, we first examined whether the transition from flexible to fixed bynames actually occurred in this period. Before 1730, 42% of birth records included a paternal byname. After 1880, this proportion had increased to 89%. We also observe a clear east/west gradient in these data, which changes over time, as shown in Table 3. To assess significance, we computed the proportion of the births for which a paternal byname was present, in each parish, in each time period. Then we fit a mixed-effect linear regression model predicting this proportion based on the longitude, binned birth year (coded 1 to 5), and their interaction. We included a random intercept for parish, with a random slope for birth year, following a maximum appropriate random effect structure from the nature of the dataset, as recommended in[42]. We found significant main effects of longitude (farther east means more paternal bynames; two-tailed $t(381.8) = 8.758$, $p < 0.001$, $\beta = 0.073$, 95% confidence interval=[0.057, 0.090]) and birth year (more paternal bynames overall as time progresses; two-tailed $t(350.4) = 5.505$, $p < 0.001$, $\beta = .259$, 95% confidence interval=[0.167, 0.351]), and an interaction between them

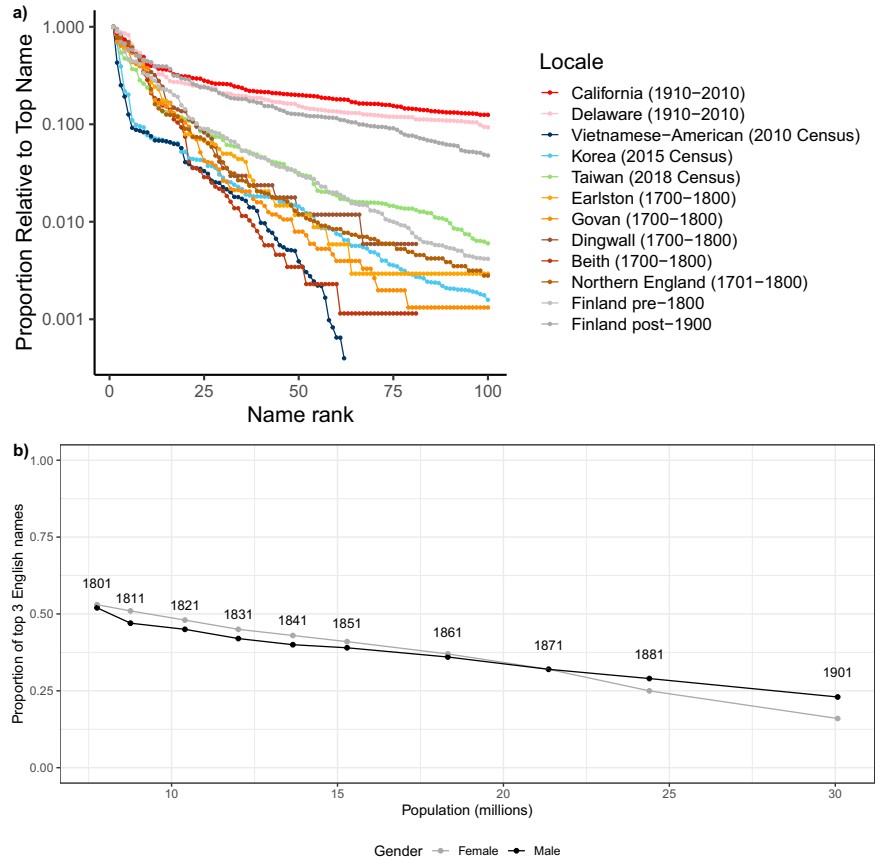

**Fig. 2 | Frequencies of most frequent names across places and time. a** The 100 most frequent prefix-names by rank in the prefix-name distributions analyzed in testing hypothesis 1 (plotted as a proportion of the most frequent name): four parishes in Scotland, 1701-1800 (Beith, Dingwall, Earlston, and Govan), two counties in Northern England (Durham and Northumberland) between 1701 and 1800, Finland pre 1800 and post 1900, South Korea (2015 census data), Vietnamese-American (reconstructed from US census 2010), Taiwanese (2018 census data), and US names between 1910 and 2010 from Delaware and California. The *y*-axis is logarithmic. Critically, before the introduction of naming laws, the prefix-name distributions of the Scottish parishes, Northern England, and pre-1800 Finland are highly skewed, and similar to those of Taiwanese, Korean, and Vietnamese-American prefix-names. By contrast, after laws required people in the West to inherit their bynames, these distributions began to change as the information conveyed by the prefix-name increased, as revealed by the longer, flatter tails of the modern prefix-name distributions in California, Delaware, and post-1900 Finland. **b** The frequency at which the top three male and female prefix-names were given in England by decade 1800–1880 and 1900. As can be seen, these frequencies declined as the population grew, which is again consistent with our prediction that fixed bynames and population growth will be associated with flatter name distributions, and concomitant increases in the information conveyed by prefix-names (data from Table 1 in ref. 71, plotted against the total population at each time point).

(which we take to mean that the effect of longitude gets smaller as birth year increases; two-tailed $t(352.4) = -3.913$, $p < 0.001$, $\beta = -0.001$, 95% confidence interval=$[-0.011, -0.004]$). We also fitted the data with a null model without the interaction term between longitude and birth year, and we found the model with an interaction term explained the data significantly better than the model without one (one-tailed $\chi^2(1) = 15.059$, $p < 0.001$).

We then examined how the adoption of hereditary fixed bynames affected the informativity of prefix-names. We expected that prefix-name entropy would increase along an east/west gradient, initially increasing more in the east than the west. Figure 3 projects the entropy of prefix-name name distributions on a map of Finland. Each panel is a different time period and, moving from left to right, reveals an overall increase in prefix-name entropy modulated by an east/west cline that becomes less prominent over time.

Our results are not dependent on the particular sample presented here. In Supplementary Note 1, we repeat these random sampling procedures and analyses 500 times, and show that the results are robust.

Having established the east/west patronymic-byname use effect, which starts large and decreases over time, we then asked whether prefix-name entropy shows a similar pattern. We ran a mixed-effect

regression predicting prefix-name entropy, calculated by sampling 50 births per bin, as described above. The predictors included binned time period, longitude, and their interaction. We also included random intercepts for parish, and a random by-binned time period slope for parish. There were main effects of birth year (more prefix-name entropy over time; two-tailed $t(301.5) = 4.591$, $p < 0.001$, $\beta = 0.256$, 95% confidence interval=$[0.146, 0.365]$), longitude (more prefix-name entropy in the east than the west; two-tailed $t(280.2) = 6.475$, $p < 0.001$, $\beta = 0.044$, 95% confidence interval=$[0.031, 0.058]$), and crucially a significant negative interaction (two-tailed $t(301.5) = -2.860$, $p = 0.005$, $\beta = -0.006$, 95% confidence interval=$[-0.011, -0.002]$; by a likelihood ratio test comparing the full model to an identical model without the fixed effect interaction term, one-tailed $\chi^2(1) = 8.053$, $p = 0.004$).

We also examined the Spearman rank correlation between the proportion of names in a parish with a paternal byname and the prefix-name entropy in that parish. In each time period, the correlation was positive, and it steadily decreased over time as shown in Table 4. These results suggest that the fatter-tailed distribution of modern prefix-names results from the changes in the information structures of names resulting from the introduction of hereditary fixed bynames.

**Table 2 | Birth year categories for Finnish data, with counts**

| birth year bin | count |
| --- | --- |
| up to 1730 | 214617 |
| 1730-1780 | 950686 |
| 1780-1830 | 1745407 |
| 1830-1880 | 1549536 |
| post-1880 | 448441 |

**Table 3 | Percentage of births registered with a paternal byname**

| | Birth Year | East | West |
| --- | --- | --- | --- |
| 1 | up to 1730 | 72% | 29% |
| 2 | 1730–1780 | 67% | 37% |
| 3 | 1780–1830 | 71% | 45% |
| 4 | 1830–1880 | 81% | 57% |
| 5 | post–1880 | 94% | 79% |

Patronymic-byname use in the East starts high and increases from 72% to 94% in the period 1700–1900. By contrast, this increase is much sharper in the West, where it is 29% at the beginning of this period and increases to 79% by its end.

## Study 3: Mixing of naming systems with different conventions

The information in U.S. prefix-names grew exponentially in the 20th century[43], whereas EACS prefix-name distributions hardly changed[25]. The effect of this is easily quantified: the prefix-names of the 49.7 million South Korean citizens in the 2015 Census have an entropy of 4.82 bits; for the prefix-names of the 27 million Californians registering for Social Security between 1910–2010, 10.16 bits (Table 1). Since entropy is logarithmic, in theory this means that the information processing cost of Californian prefix-names is now about 40 times more (≈ 5.3 bits more information) greater than Korean prefix-names, whereas our earlier analyses indicate that 200 years ago, Korean and English prefix-names contained roughly the same amount of information. This allows us to make detailed predictions about the cross-cultural impact of using Western naming conventions as a gold standard for indexing in science and publishing.

Before describing these predictions, it is important to stress how different ways EACS and Western names were legalized can make comparing them confusing (and lead to confused remarks like, "[in China] they put their last names first"[44]). This confusion arises because in the EACS, prefix-names are fixed and inherited, and bynames are 'given', whereas in the West, prefix-names are 'given', and bynames are fixed and inherited (see Fig. 1). However, for current purposes, what is important is that both Western (fixed patronymic) bynames and EACS (given) bynames are far more diverse and more

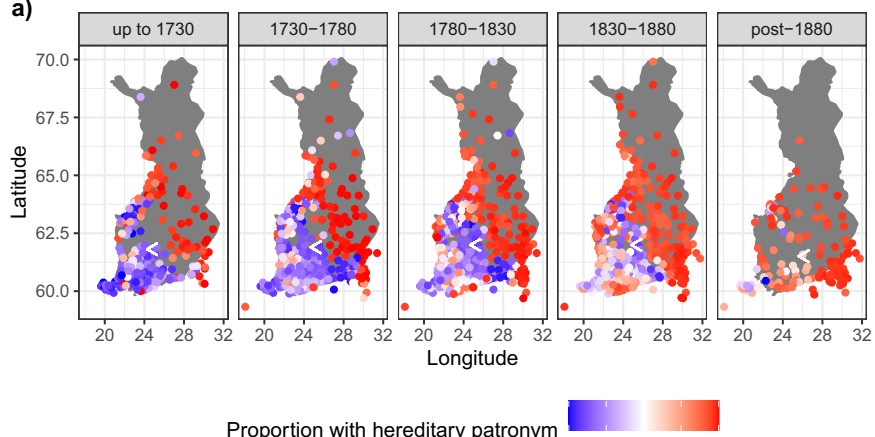

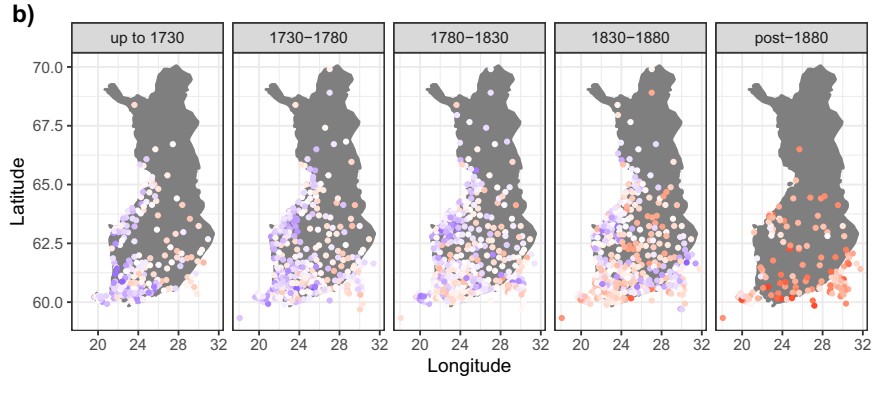

**Fig. 3 | In Finland, prefix-name entropy increased as the government required residents to have a hereditary patronym. a** Proportion of birth records in each Finnish parish at each time period containing a hereditary byname. **b** prefix-name entropy (for 50 randomly sampled names per parish) in each parish in each period. The plots indicate that Finnish prefix-name entropy increased as a proportion of the increased use of hereditary bynames as the government imposed name regulations, both of which are reflected in an east/west gradient of change: the proportion of hereditary bynames (**a**) and increases in prefix-name entropy (**b**) are initially evident more in the east than the west, before slowly spreading westwards across the country.

**Table 4 | The Spearman rank correlation between proportion of patronymic-bynames present and the prefix-name entropy, between proportion of patronymic-bynames present and longitude, and between prefix-name entropy and longitude, along with their 95% confidence intervals (CIs) and p-values**

| Birth Year | Variable Pair | Spearman $\rho$ | 95% CIs | *p*-value |
|---|---|---|---|---|
| up to 1730 | Patronym:PrefixNameEnt | 0.29 | (0.16, 0.41) | <0.001 |
| | Patronym.:Longitude | 0.54 | (0.43, 0.65) | <0.001 |
| | PrefixNameEnt:Longitude | 0.49 | (0.38, 0.60) | <0.001 |
| 1730–1780 | Patronym:PrefixNameEnt | 0.22 | (0.11, 0.32) | <0.001 |
| | Patronym.:Longitude | 0.35 | (0.24, 0.46) | <0.001 |
| | PrefixNameEnt:Longitude | 0.40 | (0.31, 0.49) | <0.001 |
| 1780–1830 | Patronym:PrefixNameEnt | 0.18 | (0.08, 0.27) | <0.001 |
| | Patronym.:Longitude | 0.38 | (0.28, 0.47) | <0.001 |
| | PrefixNameEnt:Longitude | 0.21 | (0.11, 0.30) | <0.001 |
| 1830–1880 | Patronym:PrefixNameEnt | 0.10 | (0.003, 0.20) | 0.039 |
| | Patronym.:Longitude | 0.42 | (0.34, 0.51) | <0.001 |
| | PrefixNameEnt:Longitude | 0.01 | (-0.08, 0.11) | 0.77 |
| post-1880 | Patronym:PrefixNameEnt | 0.06 | (-0.13, 0.24) | 0.52 |
| | Patronym.:Longitude | 0.72 | (0.63, 0.81) | <0.001 |
| | PrefixNameEnt:Longitude | 0.16 | (−0.01, 0.34) | 0.058 |

The CIs are calculated using the SpearmanCI package[69] in R[70].

**Table 5 | The prefix-name entropy and the byname entropy of matched samples taken from 2550 scientists from 6 different countries**

| | Country | Prefix-name Entropy | Byname Entropy |
|---|---|---|---|
| 1 | China | 6.15 | 8.64 |
| 2 | Finland | 6.94 | 8.54 |
| 3 | France | 7.03 | 8.68 |
| 4 | Korea | 4.72 | 8.71 |
| 5 | Russia | 5.27 | 8.58 |
| 6 | US | 7.43 | 8.65 |

informative than US (given) prefix-names and EACS (fixed patronymic) prefix-names (See e.g., Table 5).

Because prefix-names convey less individuating information than bynames (and the individuating information in initials is similarly constrained), it follows: (1) Under current indexing conventions, which combine initialized 'given names' with full inherited names, EACS names will convey far less individuating information than Western names, making EACS names more confusable; (2) If indexing conventions were recast to reflect the information structure of name phrases, the confusability difference between Western names and EACS names could be drastically reduced. Note that in this study, we are not analyzing names under the categorization of prefix-names and bynames as in previous ones. Instead, we will follow the (Western) convention of name categorization and the indexing convention arising from it to show how such conventions lead to greater confusion if it is applied to EACS names. To avoid confusion, in this study, we will refer to the part of the name that is given as "flexible given name" and the part of the name that is inherited as "fixed inherited name."

To test the hypotheses, we obtained cross-cultural samples of scientific communities by scraping membership lists of academic/scientific societies in the United States, China, Korea, France, Finland, and Russia[45–50] (see Study 3 in methods). We explicitly chose these samples in order to examine the possible real-world consequences of current indexing conventions (and hence the transliteration of character-based Chinese and Korean names to the Latin alphabet) in a domain where the communication of identities is critical, namely academic publishing.

The lists of names were then spot-checked, and typos or extraction errors were fixed as needed. The exact list of names used is in the repository associated with this paper. To compare across datasets, we randomly downsampled all datasets to make them the same size as the Korean set (425 individuals), and then computed the number of repeated names: for example, if "Robert" occurs 4 times, its repeat count is 3. This was done for each: flexible given name, fixed inherited name, full flexible given name + fixed inherited name initial, and flexible given name initial + full fixed inherited name (see the top left panel in Fig. 4). We then generated sets of possible shortened versions of the names by taking only the flexible given name, only the fixed inherited name, or the abbreviated initials of either the flexible given name or the fixed inherited name.

We then measured various properties of these distributions, which generated the following name styles: flexible given name initialized (e.g., C. Darwin / Z. Li), fixed inherited name initialized (e.g., Charles D. / Zhaoping L.), flexible given name alone (e.g., Charles / Zhaoping), fixed inherited name alone (e.g., Darwin / Li).

We also considered the potential influence of two further naming conventions present in these samples. The first is the presence of Korean generational names, which (in South Korea at least) typically comprise the first syllable of a flexible given name, which is shared across siblings, such that the second syllable is unique to the individual, e.g., Korean soccer-playing brothers Son Heung-min and Son Heung-yun. Accordingly, in Korean, names are typically indexed by both syllables, such that Son Heung-min's name is rendered as "H M Son". However, in our basic analysis the given name of Hyun Jae in Fig. 4 is simply rendered as H. To better reflect the specific structure of Korean names, we also explored an alternative rendering that treated each syllable in a Korean given name independently, such that the given name initial of Hyun Jae was represented as H. J. instead of H. We denote this alternative convention as "Korea-2" in Fig. 4.

The second alternative consideration relates to patronyms in Russian names. Unlike most Western societies, a full Russian name also includes a third component in addition to flexible given name(s) and a fixed inherited name: a true patronym, derived from a parental given name (a practice also common in other East Slavic and ex-Soviet societies). For instance, the late Russian linguist Vadim Kasevich's full name is Vadim Borisovich Kasevich, reflecting the fact that his father was Boris, and the late linguist Alexandra Kamynina's full name is

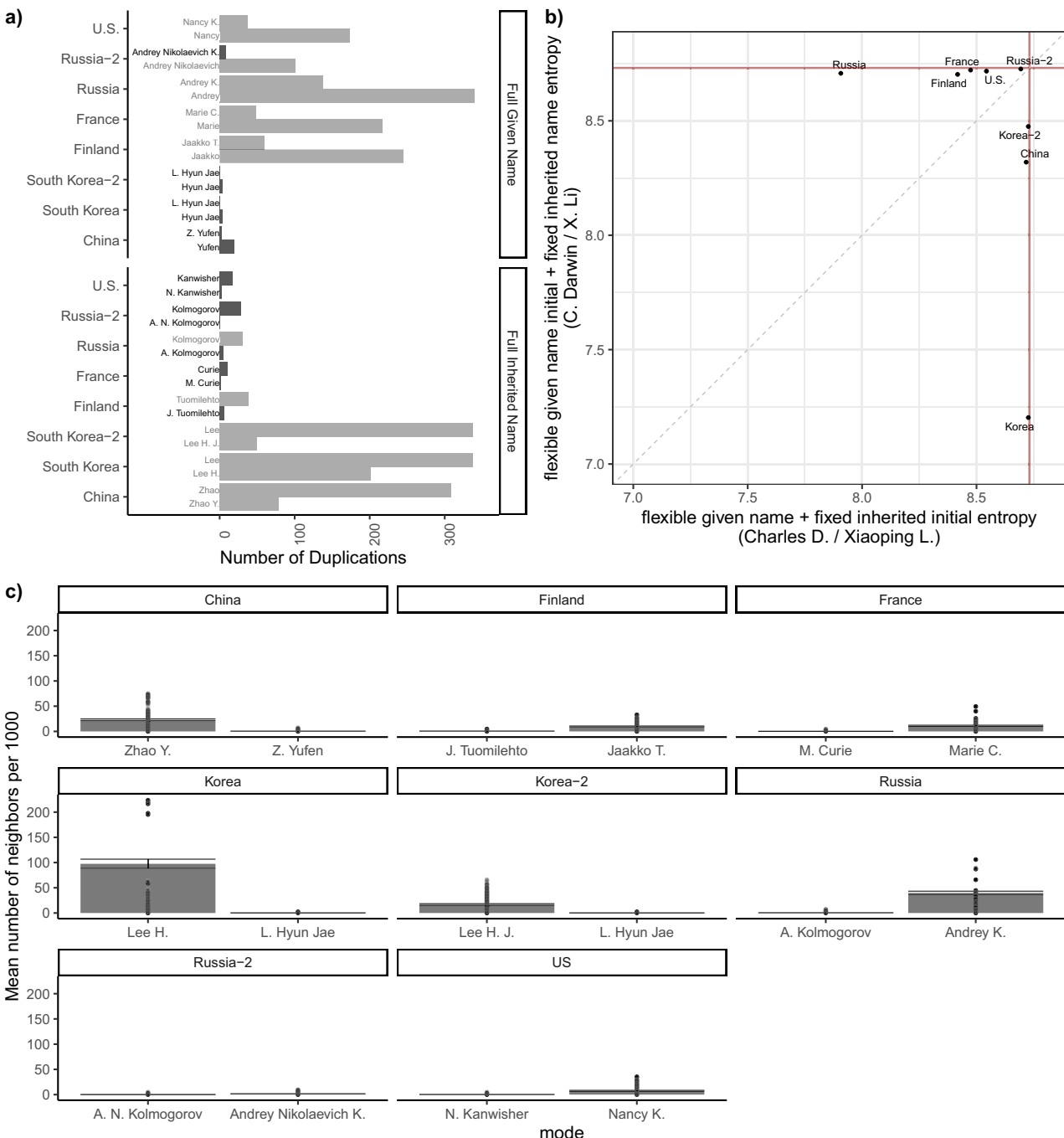

**Fig. 4 | Imposing the indexing convention from one culture to another makes name indices less informative and more confusable. a** Number of repeated names, for each of flexible given name, fixed inherited name, and the two initialism conventions we consider (i.e., Russia-2 and South Korea-2). The darker bars show naming conventions which are mostly unambiguous on the scale of scientific communities as measured in our sample. Russia-2 and Korea-2 refer to forms using the Russian patronymic and Korean generational name as part of the initials (e.g., A.N. Kolmogorov and Lee H. J.). **b** Entropy of names of the form C. Darwin vs. Charles D., across cultures. The dark red line represents the maximal attainable entropy from 425 names. **c** The likelihood of confusion, as measured by the mean number of neighbors per 1000 for each name style (left: flexible given initial + fixed inherited name; right: flexible given name + fixed inherited name initial) among scientist names in each country. Error bars represent 95% bootstrapped confidence interval. Two-tailed t-tests suggested that EACS names are more confusable than Western names when they are indexed by flexible given initial + fixed inherited name ($t = 24.014$, $p < 0.001$), and Western names are more confusable than EACS names when they are indexed by flexible given name + fixed inherited name initial ($t = -28.531$, $p < 0.001$).

Alexandra Alekseevna Kamynina, since her father's name was Aleksey. Russian publications often cite individuals by their flexible given name initial, their patronymic initial, and their full inherited name, rendering these two linguists as V. B. Kasevich and A. A. Kamynina, respectively. This alternative convention is denoted as "Russia-2" in Fig. 4.

Our analysis also makes clear that current academic publishing conventions (flexible given name initial + full fixed inherited name) lead to more ambiguity for Chinese and Korean names than for U.S. names, even though almost every full name phrase was unique in every dataset. Figure 4 plots the entropy of the flexible given name + the fixed

inherited name initial (i.e., where Charles Darwin, Charles Dickens, and Charles Dodgson are all indexed as Charles D.) against the flexible given name initial plus the fixed inherited name (C. Darwin, etc.), and shows that when EACS names are *not* initialized according to Western conventions they are far less ambiguous. In addition, indexing every syllable in a Korean flexible given name (e.g., H. M. Son instead of H. Son) increases the entropy of the index distribution dramatically, and initialized Russian inherited names become less confusing if the initials for both the full flexible given name and the full patronyms are utilized.

In our analysis so far, we have treated each name atomically, but it cannot capture a critical factor in the empirical functioning of name systems: psychological confusability. Words are harder to process in the presence of competitors, and this effect increases as discriminability decreases[51]. To estimate the effect of this factor, we computed the number of orthographic neighbors in our datasets, defining neighborhoods in terms of the number of 0-edit and 1-edit neighbors of each name under the possible initial + name sequences, namely flexible given name initial + full fixed inherited name, and full flexible given name + fixed inherited name initial. The results are illustrated in Fig. 4. EACS names have many more 1-edit orthographic neighbors when indexed in the Western convention (flexible given name initial + full fixed inherited name; two-tailed $t(1274.3) = 24.014$, $p < 0.001$, $\beta = 19.309$, 95% confidence interval=[17.732, 20.888]): Chinese names have an average 22.79 neighbors per 1000 names, and Korean names have an average of 97.5 neighbors per 1000 names, compared with Western countries where the numbers are lower than 0.5 per 1000 names. In addition, similar to the results in the entropy analysis, the number of 1-edit neighbors is sharply reduced for Korean names under the Western indexing convention when every syllable in a Korean flexible given name is indexed (16.7 per 1000 names, compared to 97.5 if only the first syllable is indexed). On the other hand, EACS names have many fewer 1-edit neighbors than Western names when the names are indexed by full flexible given name + fixed inherited name initial (two-tailed $t(2134.1) = -28.531$, $p < 0.001$, $\beta = -5.813$, 95% confidence interval=[−6.212, −5.413]). In our sample, "H Kim" is the index most likely to be confused, with 95 1-edit neighbors such as W Kim, B Kim, and H Lim. "Vladimir M" is the most confusable index in non-EACS names, having 45 1-edit neighbors such as Vladimir T and Vladimir D. But when including the full patronym, Russian name indices become much less likely to be confused.

## Discussion

In the sample examined above, Western naming conventions resulted in far more confusable and ambiguous renderings of EACS names. To put these results in a wider perspective, in 2010, neuroscientist Li Zhaoping described how the conventional way of publishing her name in U.S. academia (Z. Li) made it ambiguous[52]. Given the importance of name recognition and name memorability in academia[53–56], Zhaoping went on to describe how the confusability that results from the way their names are rendered in western publishing continuously puts EACS scientists at a disadvantage, and announced that while by convention her name ought to written as 'Z. Li' for scientific publishing, she had decided to use L. Zhaoping instead.

Our analysis provides quantitative support for this position by showing that when the names of all of our EACS scientists are encoded Zhaoping's way, duplication rates in the EACS and Western datasets become similarly low. These results can enable other scientists to make decisions about their authorial identities in the context of a broader theoretical and historical framework that provides a more complete account of the functionality and effectiveness of names.

We suggest that increased popular awareness of these issues can play an important role in increasing cross-cultural understanding. The common misconceptions currently associated with name codes can make the world's name systems appear arbitrary. Researchers examining Chinese given names emphasize how "individuality is

expressed by the [Chinese] given name" and note how "the opportunities for creativity and originality in the [Chinese] given name are much greater than in English name-giving tradition"[44]. Meanwhile, Western-focused narratives make exactly the 'opposite' point when it comes to the expressiveness of inherited names: "how tedious it would be if British surnames had the same neatness - and terseness: Park is a longer than average name in Korea. Koreans don't have names which ramble away into the distance like Featherstonehaugh, Haythornthwaite, or McGillicuddy"[57]. We have shown how, when seen from an information-theoretic perspective, the world's name systems share a common structure. In the languages examined here, a relatively small set of largely meaningless prefix-names was used to mark other lexical items as bynames to yield systems that, at least historically, enabled large sets of identifiers to be created without inflating the lexical inventories of languages. These vernacular name systems allowed identities to be communicated in ways that were smooth and efficient.

We have also shown how modern differences in these systems arose out of the different ways in which vernacular names were repurposed as fixed identifiers. And we showed how the lack of attention to the false equivalences that these changes have created in name systems can lead to unforeseen negative outcomes—along with some simple ways in which, once the common information structure of names is understood, these negative outcomes could be reduced or even eliminated.

If we extrapolate from historical data anywhere in the premodern world, 12–15 of the main characters in *Game of Thrones* ought to have shared the same most common prefix-name. Instead, the prefix-name of almost every character was unique! What our results indicate is that, while this kind of unique name system might seem plausible in the realm of fantasy, historical name systems were very different and featured large amounts of duplication. Information theory not only provides an explanation for that duplication, it can help explain why name systems across languages shared a common structure in the past. It also explains how important it is to understand this structure when it comes to manipulating these systems, whether this be for the purpose of creating official identifiers, or establishing standards in publishing. While accepting that the requirements of indexers must also be considered, these results indicate that a reversion to something closer to the historical norm – allowing individuals more flexibility in choosing how their names are presented – not only has strong historical and information-theoretic precedent, it could both help individuals reduce nominal ambiguity and do so more efficiently in the future.

## Methods

This study was reviewed and declared exempt by the Institutional Review Board at the University of Texas at Austin as STUDY00004122.

### Study 1

To estimate and compare prefix-name entropy across cultures and historical time, we drew name data from various sources. In what follows, we describe these datasets in further detail, and characterize how we compared them. Our usage of these datasets are compatible with their respective terms and conditions (See Supplementary Note 2 for more details).

**Materials.** US Name Data were taken from birth name records made available by the Social Security Administration[38]. The dataset includes prefix-names of babies born between 1910 and 2018 in all 50 states in the US. Since 1910 was the first year when the dataset was available, and 2010 was the last year for which census name data were available, the data between 1910 and 2010 were analyzed. Only prefix-names that are shared by more than 5 people in each state in each year are included in the data, with entries being restricted to cases where an individual's

birth year, sex, and state of birth are on record, and where their given name is at least 2 characters long. The names in the records are then pre-processed to remove hyphens and spaces, so that Julie-Anne, Julie Anne, and Julieanne all count towards the same name's frequency.

For our detailed analyses, we chose a representative small state (Delaware) and a representative large state (California); however, further analyses showed that the name data for other states pattern similarly.

The majority of the Vietnamese-Americans sampled in the 2010 census[32] are naturalized citizens who were born-and named-in Vietnam[58]. This made it possible to recreate a distribution of Vietnamese prefix-names (family) from US family name data to examine whether previous findings regarding Korean and Chinese prefix-names (which are exponentially distributed[22,24]) generalized to other EACS languages (see Fig. 4a). To make this estimate, two lists of frequent Vietnamese prefix-names[59,60] were combined with a Vietnamese names list from Wikipedia. This yielded a total of 62 prefix-names (family names), the frequencies of which were then extracted from a dataset containing counts of the family names that occurred most frequently in the US 2010 census published by the US Census Bureau (source:[32]). The US census data contains family names shared by at least 100 people, along with the number of people bearing each last name, and the percentage of each name's distribution across 6 Race and/or Hispanic Origin subgroups.

US census entries use only the 26 letters of the English alphabet, which means that when many Vietnamese names are entered into it they become confusable with Western family names (e.g., the Vietnamese family name "Bach" is entered as "Bach", a typically German family name). To improve the accuracy of our estimate, we multiplied the number of people having each last name by the percentage of people with that last name in the Asian/Pacific Islanders (API) subgroup.

Name data for South Korea were obtained from the Korea National Statistic Office website, which published a dataset containing a list of the prefix-names shared by more than 5 people in 2015 census, along with the number of people with each name[30]. The prefix-names are listed in both Hangul (a phonetic writing system) and Hanja (a logographic writing system based on Chinese characters). Since multiple prefix-names written in Hanja sometimes share the same Hangul (e.g., the prefix-names in Hanja "到","度", "桃", "覩", "道", "都", and "陶" are all written as "도 (Do)" in Hangul), we use Hanja characters as our unit of analysis. There are in total 533 Hanja prefix-names in the census, covering 49.7 million people in total.

Taiwanese Name Data was taken from a list published as part of a report by the Taiwanese Ministry of Interior in 2018 (source:[31], Table 57, pp.282–304). It contained 1832 prefix-names listed using traditional Chinese characters, along with the number of people sharing each name. We manually logged the 500 most popular prefix-names, covering a population of 23.5 million. The 500th most popular prefix-name is shared by 135 people.

Finnish Name Data were extracted for the period post 1900 from a larger set of genealogical Finnish name records collected and organized by[36].

The Scottish data, was extracted from the National Records of Scotland and published as an appendix in ref. 27, and comprises the prefix-names recorded in four parishes (Earlston, Govan, Dingwall, Beith,) from 1700 to 1800. The English name data were extracted from George Bell's parish marriage register transcriptions for Northumberland and Durham between 1701 and 1800[29].

The English records had been pre-processed to take account of alternate spellings of the same name (e.g., Ann / Anne / Hanna) and common abbreviations (e.g., Thomas being abbreviated to Thos.). Accordingly, we applied the same process to the Scottish data, as well as additionally removing obvious bynames that had been mistranscribed as prefix-names (e.g., where McVey is given as a prefix name). To estimate the potential impact of this kind of pre-processing, test comparisons between the processed and unprocessed

**Table 6 | The prefix-name entropy of four Scottish parishes calculated using the original data and the pre-processed data (where we account for alternate spellings of the same name, name abbreviations, mistranscription, etc.)**

| Parish | Entropy before pre-processing (bits) | Entropy after pre-processing (bits) |
|---|---|---|
| Beith | 4.42 | 4.35 |
| Dingwall | 5.00 | 4.87 |
| Earlston | 4.85 | 4.85 |
| Govan | 4.84 | 4.70 |

The result from the pre-processed data was the one shown in Table 1.

distributions were then conducted. The results of these comparisons indicate that the effects of pre-processing on comparisons between name distributions are marginal (all pointwise $R^2 > 0.98$), suggesting that the pre-processing that many of the other datasets we analyzed were subject to would have been unlikely to significantly affect the results of our analyses. We also calculated the prefix-name entropy of the four Scottish parishes using the original data (results presented in Table 6), and we again found the influence of pre-processing on entropy estimation to be minimal.

The premodern Finnish name data are taken from records made before 1800 in the same dataset from which the modern Finnish name data were taken from[36]. The pre-1800 dataset comprises 1748523 birth records in total, with 1833 unique names, and again, spelling variations have been standardized as per[36].

## Procedure

For each location, we calculated the entropy of the prefix-name distribution as a measure of the information conveyed by the distribution. For privacy reasons, our modern datasets do not report name data below a certain population threshold. To account for these missing data, we estimated a lower bound and an upper bound of the true prefix-name entropy in each location.

As was described in the Results section, the entropy calculated from the datasets serves as a lower estimate of the true entropy. If a dataset contains $N$ names from $M_0$ people, and the occurrence frequency of name $i$ is $f_i$, then $M_0 = \sum_{i=1}^{N} f_i$, and the lower bound entropy of the dataset can be estimated by Equation (2) below:

$$H_{low} = - \sum_{i=1}^{N} \frac{f_i}{M_0} \log_2 \frac{f_i}{M_0}. \quad (2)$$

Unlike our modern samples, the pre-modern Western datasets for Finland, Northern England, and Scotland are comprehensive, such that the entropy from Equation (2) is the true entropy of the population.

For the other datasets, an upper bound estimate is calculated as follows. First, we obtain the population of each locale from census data[61]. The population of the entire US, California, Delaware, and Vietnamese-Americans is extracted from the 2010 US census data[62–65]. The population of Taiwan is taken from data published by the Taiwan Department of Household Registration in January 2018[66]. The population of South Korea is extracted from the 2015 Korean census[67]. If the population of a locale is $M$, then there are $M - M_0$ persons whose prefix-name data are left out in our datasets. We assume each of these $M - M_0$ persons gets a unique prefix-name, which makes the total number of unique prefix-names $N + M - M_0$, and $f_i = 1$ for $i \in [N + 1, N + M - M_0]$. Therefore, the upper bound entropy can be calculated as

$$H_{up} = - \sum_{i=1}^{N+M-M_0} \frac{f_i}{M} \log_2 \frac{f_i}{M}. \quad (3)$$

The discrepancy between $H_{low}$ and $H_{high}$ will generally be higher when the data are more incomplete.

## Study 2

**Materials.** As noted above, the combination of well-preserved genealogical sources and the fact that the transition from flexible to hereditary fixed bynames proceeded asymmetrically in Finland (hereditary bynames became dominant in eastern Finland earlier than western Finland[37]) mean that the Finnish name records described above[36] provide a unique opportunity for exploring the consequences of changing previously flexible bynames into the systems of hereditary fixed patronymic-bynames widespread today.

For the period 1600–1917, the dataset comprises 4908687 birth records in total, with 5377 unique prefix-names. Each birth record comprises the prefix-name of the baby, the parish where the baby was born and recorded, and, when available, a paternal byname. We divided the records into 5 different year bins: pre-1730, 1730–1780, 1780–1830, 1830–1880, and post-1880 (no data were excluded from this analysis).

**Procedure.** We focused on the following variables for each birth:
- parish: The church parish where the baby was born and recorded.
- latitude: The latitude of the parish.
- longitude: The longitude of the parish.
- baby first name (`child_first_nameN` in the original data): The first/given/prefix-name of the baby.
- father byname (`dad_last_nameN` in the original) The byname, when available, of the father. This serves as a critical variable in that we use the choice to include or exclude the father's byname as reflective of the usage of fixed, patronymic-bynames in the parish. When the father's patronym (father's name, a separate variable in the original data) is reported, we do not include this.

V We filtered to include birth years from 1600 on. That left us with data from 482 parishes, with birth years from 1600 to 1917. The birth years were broken into discrete categories, as shown in Table 2.

To compare equal-sized amounts of data, for each parish and for each birth year bin, we sampled 50 births.

From this data, we computed, for each parish and birth year group:
- the entropy over first names, computed as $\sum_i - p_i \cdot \log_2(p_i)$ (as per Study 1).
- the proportion of births that included the father's surname.

## Study 3

**Materials.** The list for U.S. scientist names were extracted from the United States National Academy of Sciences[49], an honor society for academics in the sciences, broadly construed. The list contains living and deceased members. We extracted the part of the name that comes first as the flexible given name and the part that comes last as the fixed inherited name. We treated the name that appears between as a middle name, but did not consider it since typically (although not always) middle names are not used in the U.S.

The list for Chinese scientist names were extracted from the Chinese Academy of Sciences[45], a scientific honor society in China, containing living and deceased members. The list contains members elected as early as 1955. Names on the Wikipedia site appeared transliterated (as pinyin, Latinized script) and were already displayed in the Western name order, e.g., Zhiqin Xu. Most names consisted of two parts. We assume the part of the name listed first is the flexible given name (the byname in Chinese), and the second name listed is the fixed inherited name.

The list for French scientist names were taken from the French Academy of the Sciences, a scientific honor society in France. The list[47] contains "full members, adjunct members, corresponding members, and foreign associated members, of all times, since its foundation until today." We treated the part of the name that comes first as the flexible given name and the part that comes last as the fixed inherited name.

The list for Finnish scientist names were taken from the Finnish Academy of Science and Letters contains. We copied the list of "domestic members" from the site given[46]. The list of members is made publicly available by the Academy. There were typically two elements to each Finnish name, and we treated the part that comes first as the flexible given name and the part that comes second as the fixed inherited name.

The list for Russian scientist names were extracted from the Russian Academy of Sciences, which contains names of the electees on its website[50]. The names are made publicly available by the Russian Academy of Sciences. We extracted the names "full members" from the site, which provides them in Cyrillic script. All subsequent analyses were conducted using this script. Names for Russian individuals typically included three elements: a flexible given name, a patronymic, and a fixed inherited name. We extracted the list into these three elements when possible. We consider names with and without patronymics in our analyses.

The list for Korean scientist names were extracted from the Korean National Academy of Science, which includes members from the humanities, social sciences, and natural sciences. We copied the publicly available names from the Academy's website[48]. Korean names often have a fixed inherited name and a flexible given name, where the flexible given name has two components that are separated by a space or hyphen when transliterated. In our dataset, the names are transliterated ahead of time. When the flexible given name is separated by a space or hyphen, we conduct two different analyses: treating it as a single given name or as two components. We make this decision since some Korean writers publishing under Western conventions use a double-barreled initialism with a full fixed inherited name (e.g., K.W. Chang for Chang Ki Won).

Three of our samples were obtained from Wikipedia, copied in a way consistent with Wikipedia's terms of service. The other 3 samples (for Korea, Finland, and Russia) were copied from the websites of their respective academies. In all cases, the names of members are intentionally provided to the public as public information and were accessed as intended.

**Procedure.** We used these datasets to obtain cross-cultural samples of names from the relevant communities. These were then processed and downsampled as described in Study 3 Results.

### Reporting summary

Further information on research design is available in the Nature Portfolio Reporting Summary linked to this article.

### Data availability

The American, Korean, and Taiwanese name census data, as well as the birth records of parishes in Scotland and northern England, have been deposited on Zenodo (https://doi.org/10.5281/zenodo.17644337)[68]. The Finnish birth records are available from ref. 36. The scientist name data are not available due to data privacy concerns, but can be accessed on request by contacting the corresponding author.

### Code availability

Code for reproducing analyses is available on Zenodo (https://doi.org/10.5281/zenodo.17644337).

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

## Acknowledgements

Some parts of this manuscript are based on work previously presented at the 35th Annual Conference of the Cognitive Science Society and published in its Proceedings. We thank Amalia Spyromilio, Stella Schneider, Holly Jenkins, Evelina Fedorenko, Ted Gibson, and Damian Blasi for helpful comments. M.R.'s contribution was supported by a grant from the Deutsche Forschungsgemeinschaft (DFG 547529231). We acknowledge support from the Open Access Publishing Fund of the University of Tübingen.

## Author contributions

M.R.: conceptualization, data curation, formal analysis, investigation, methodology, validation, visualization, writing—review and editing. S.C.: conceptualization, data curation, formal analysis, investigation, methodology, software, validation, visualization, writing–review and editing. R.F.: conceptualization, data curation, writing—review and editing. K.M.: conceptualization, data curation, formal analysis, investigation, methodology, validation, visualization, writing—review and editing.

## Funding

## Competing interests

The authors declare no competing interests.
