## [Transparent Peer Review File · Nature Communications]

Cross-Cultural Structures Of Personal Name Systems Reflect General Communicative Principles

Corresponding Author: Dr Michael Ramscar

Version 1:

Reviewer comments:

Reviewer #1

(Remarks to the Author)

Review report for "Cross-Cultural Structures Of Personal Name Systems Reflect General Communicative Principles" by Ramscar and Mahowald.

This paper studies the structures of personal names in several different cultures and identifies some general patterns in personal-name codes. By analyzing several datasets of names, including US Baby Name Data, Vietnamese names, South Korean names, Scottish and Northern England data, and others, the authors find two simple principles underlying name systems: (1) It should encode large numbers of identities, and (2) it should ensure these encodings are usable. Interestingly, the authors use the implementation of patronyms to identify the change in Western name distributions, and they further study potential bias against individuals from some cultures.

The personal name is often a reflection of culture and human society in general. From a computational social science and the science of science perspective, I found it interesting to study cross-culture differences in name systems or the information structure of names using large-scale datasets, as it provides a unique angle for us to understand cultural evolution across a much longer period of time. At the same time, I feel it remains challenging to interpret these empirical observations, especially when it comes to the claim of the systematic bias against individuals with names from non-Western cultures. In the following, I would like to provide some comments and suggestions, hoping to help the authors to develop this study further.

1. Personal name is a product of human language, which is more often a local (original) language than widely used English. The translation of names from the original language to English may substantially affect the analysis in this study. In the example illustrated in Figure 1, the movie director Woo Yu-Sen (吴宇森), aka John Woo, was born in Guangzhou, China (at that time ROC) and grew up in Hong Kong. His English name Woo Yu-Sen is a translation from Chinese to English, which is likely following Wade-Giles romanization at that time and in Hong Kong. Nowadays, in China (mandarin in mainland China), people usually use Pinyin to do the translation. In this case, it would be Wu Yu-Sen. Here, Wu is the same as Woo (originally in Chinese), but the different translations result in different English words. I am wondering how this might affect the patterns the authors observe from the data, especially considering the translation from different languages.

2. In some cultures, language evolves over time, and this may also affect personal names. I understand the authors used several large-scale data across different time spans, but among them, there are some recent census data. To the best of my knowledge, initially, Korean and Japanese languages are very similar to the Chinese language due to immigration from China, while there are some legislature's implementations that drive the evolution of their language, which is basically making each word very simple in its writing. This results in the challenge that, even though the pronunciation is the same, the original words in their language are very different. In other words, there is a one-to-multiple mapping. This may significantly affect the results in Figure 2, from which we can see that Korean names are concentrated on the top-ranked ones (blue line). I am wondering how language evolution may explain or affect the authors' observations.

3. In Figure 2 and others, the authors used entropies to capture the distributions of different names. First, I would encourage the authors to elaborate their measures further. For example, how the entropy measure is calculated and whether using all names or just the top 50 names. Second, I am wondering if other measures would be more suitable to calculate the distributions of names, for example, the Gini index.

4. The authors predicted that, during the transition period, Western prefix-name information entropy would have increased as a function of the degree to which fixed patronyms were used. In Figure 3, the authors showed the frequency at which the top three male and female prefix-names were given in England by decade 1800 — 1880 and 1900. If I understand correctly, one would expect that the entropy of name distributions would increase and the Gini index would decrease after fixed patronyms were used. Therefore, it would be better to show some results on temporal trends instead of plotting the proportion against the population. I may misunderstand this, but I would encourage the authors to elaborate more on this plot and result.

5. The authors argued that “current academic publishing conventions (full hereditary-patronym, initialized bynames) lead to more ambiguity for Chinese and Korean names than for U.S. names”, and “We have shown how this potential biases against individuals with non-WEIRD names”. In my view, the evidence the authors present may not be enough to support the claim on the biases against individuals with non-WEIRD names. First, the translation of names from the original languages to English already reduces the information about names. This may have a large effect than academic publishing conventions. Second, scientists may not be a random sample from the general population, meaning that it may be hard to conclude there are biases against individuals. Third, there are very few journals using the initials of names, primarily from the physics community (e.g., high energy physics; they usually have many authors on a signal experimental paper).

6. The authors are encouraged to structure the paper according to the journal to which they submit it. Also, the resolution of figures should be enhanced.

Reviewer #2

(Remarks to the Author)

This paper observes that in Eastern Asian Cultural Sphere (EACS) pre-fix names were fixed resulting in low entropy. In old Western records, pre-fix names are also relatively low entropy; whereas, in modern Western records, pre-fix names are higher entropy. The paper argues that this change in Western cultures is due to fixing by-names, which would require high pre-fix name entropy in order to establish context-free reference. To support this, they provide data from Finland as a case study; where, as the percent of by-name fixing increases, the pre-fix entropy also increases. The paper also conducts an analysis to demonstrate that Western methods of indexing names optimizes the informativity of Western names to the detriment of EACS names. Further, there exists indexing methods that optimize the informativity of EACS names to the detriment of Western names. Therefore, we should allow for more flexible naming so as to improve communicative efficiency in-light of inefficient indexing norms (and the social good of allowing name changes). Overall, I am underwhelmed by the contribution of the paper, even though it appears technically sound.

With regards to the first part of the paper, the predictions/evidence are correlational. Are there identifiable causal pathways for how fixing a by-name increases the entropy of the pre-fix name? Are there no alternative hypotheses as to why pre-fix entropy might increase over time? For example, before patronym's were fixed did people not reuse family names or names associated with a trade (Dread Pirate Roberts) until the population increased to require other names? Does the point in the paper about it not being linked to population size because Korea has a higher population than Delaware confound average population size when patronymic names were fixed? Without these considerations/discussions of alternatives the argument sounds like a just-so story.

One point of contention: “An information-theoretic analysis cannot capture one critical factor in the empirical functioning of name systems: confusability.” I'm not convinced this is true. It might just be imprecise writing but noisy channel coding would prefer dispersion of the form; whereas, source coding would prefer similar meanings to be compressed to similar forms for optimal meaning recovery. The charitable would argue that context further helps disambiguate. Either way confusability is an important concept to an information-theoretic analysis.

With regards to the second part of the paper, I'm all on board with flexible names for the social reasons. We should, as a community, rethink using indexing systems that disadvantage large chunks of the world. That said, the subtext of the paper, as I read it, is that the predominant function of naming is establishing reference and the legal requirement for context-free reference might not be that important because we would still be communicatively efficient if we had contextualized naming (as in the past). This argument doesn't address the non-reference goals that might add functional pressures on naming systems to become context-free--e.g., inheritance, ownership and provenance. I bring this up not to undercut the authors' intention but to ask for elaboration on what kind of flexibility they are asking for and what the implications would be beyond reference.

Minor notes:

Pg 3, The predictions for the third hypothesis are vague compared to those actually tested.

The main text before Figure 2 does not introduce the Northern England dataset.

In Figure 2, it would be helpful to have different markers for EACS and Western cultures.

At first mention, it's unclear what's meant by indexing and why it's important. The examples clear it up though.

Pg 7 not initialized → initialized

The K2 and R2 conditions in Figure 5 need to be explained better.

Reviewer #3

(Remarks to the Author)

This is an interesting paper which takes an information-theoretic perspective on naming conventions and provides analyses of name data (from some rather creative sources) making 3 main points:

1. There are similarities between EACS naming systems and “pre-modern” western systems, prior to the relatively late Western standardisation that individuals’ bynames were hereditary, in that they show similar entropy over prefix names prior to the standardisation of bynames.
2. The switch to fixed by-names in Finland leads to a compensating increase in entropy in 1st names, which has an expected temporal and geographic dynamics.
3. Since EACS and western cultures order given and family names differently, there is a systematic identifiability penalty for initialising given names but not family names which disadvantages EACS authors in making them less uniquely identifiable.

I really liked the paper and would like to see it published. I did have one methodological concern, and then a couple of points of confusion that could be cleared up with a bit of an edit.

The substantive concern: regarding the low entropy of pre-modern Western prefix names, I can sort of buy that population size not the only factor here, since you take the 50 most frequent names in each data set and also, as you point out, population size doesn’t correlate perfectly with entropy. But I did wonder about other features that might artificially suppress the prefix name entropy of the smaller data sets. For instance, what about relatedness/non-independence in the Scottish samples - to what extent are these all date from a small set of families in those towns/villages? It’s old fashioned these days (like 1-2 generations out of date), but it’s not uncommon in the UK to have children take the same first name as a parent or parent’s sibling - my dad is named after his dad, the same is true of several of my friends, I think this used to be very standard and quite strictly applied, with middle names used to individuate (and people in practice being referred to by their middle name). Even if people are not following this old tradition, they may copy names they see around and like - I have the same name as my uncle, my daughter has the same name as my sister-in-law, apparently for this reason. Furthermore I think the tradition of naming sons after fathers, or copying names from elsewhere in the family, is itself a tradition that varies by families. You also get weird naming clusters (e.g. weirdly loads of kids named “Kai” in my son’s class) which I think must reflect independent name copying from a locally prominent exemplar. Those sort of effects should suppress name variation, and could (I think) have larger effects in smaller populations, e.g. where everyone is copying an already-small pool of names or naming practices. Can you discount those sorts of effects as leading to misleading parallels in prefix name entropy, e.g. in Figure 2?

Points of my confusion where a little clarification would help:

Where you say “Information theory shows how combining codewords of different frequencies (John, Claude) with other codewords conditioned on them (Smith, Shannon) can allow probabilistic codes that near optimally balance the competing demands of encoding individuals and communicating identities to be defined” - I think it would be nice to illustrate this somehow since an intuitive understanding of why this is the case would be helpful. In figure 1, I also wasn’t sure if the lower-information word should come first by design, or if the increase in information in later words is because of conditioning/combinatorics.

Regarding the increase in prefix name entropy in Finland - wouldn’t you also predict a decrease in entropy in bynames in the same period? I think that’s what figure 4 upper is intended to show, but it’s slightly indirect in that it shows instead use of father’s hereditary patronymic. Can you show the decrease in byname entropy directly, or is there a reason why this can’t be done? I wondered if part of the problem was that people were simply not using bynames in the pre-modern data - but in that case, how was the individuating function of names achieved?

I think the section on the problem with current conventions around initialising given names is really interesting - but I got quite confused here, because you bring in the terminology “Family” and “Given” names, and then (I think) what you are calling the prefix name for EACS authors is used second in scientific citations - e.g. Woo Yu-Sen becomes Y. Woo rather than W. Yu-Sen (which would actually be preferable for individuation purposes, if I am reading your results correctly). There are places where you also seem to mix the two different terminologies, i.e. describing the convention as “full hereditary-patronym, initialized bynames” - isn’t it “full hereditary patronym, initialised given name” or “full hereditary patronym, initialised byname for westerners, initialised prefix name for EACS individuals”? I also wasn’t sure what I was supposed to take from the right-hand panel of figure 5. Anyway, my uncertainty around the terminology and what I was to take from Figure 5 made me doubt that I actually had got your conclusion correctly - I *think* you are saying that e.g. while C. Darwin works nicely for WEIRD names, something like W. Yu-Sen would be more individuating for EACS names (and Y. Wu is very bad for them).

Two minor additional quibbles on figures: in figure 2, it would be better to use less code-internal variable names in the legend, and definitely not use CA and DE abbreviations, which will be much less familiar as US states to non-US readers; in figure 3, please don’t use scientific notation in the x-axis - just say “Population (millions)” and give easier-to-read numbers.

That’s it. As I said, if I am following it correctly, I think the paper is really interesting and I’d like to see it published if it can be revised to 1) spare the reader some of the confusions I had and 2) deal (in supplementary materials I expect) with my

concern re. the effects of name copying in small populations.

Reviewer #4

(Remarks to the Author)

I think this study has an enormous potential, given that it discusses a topic that is rarely analyzed with quantitative approaches, but which the authors show to have an impact with respect to the naming practice in publications in a globalizing world.

In its current form, however, the study loses a lot of its potential due to the way the results and the study are presented. It is very difficult to follow the reasoning and to understand what studies were undertaken, since the authors do not structure their study in a traditional way that would also be common for readers of the journal. This means, the authors should -- if they follow my suggestion -- completely re-do their main structure of the article, by separating materials and methods from analyses, and stating hypotheses clearly in the introduction and then discussing their findings in the conclusion. In the current form, the paper may be interesting for journals with another peer group, but the journal where the authors submitted typically looks at particular findings done by applying research methods, which need to be presented in such a structured way.

Furthermore, I miss examples in figures or boxes or tables of how name practices differ. I think it would be easy to come up with some nice illustrations, maybe even taking world maps where one plots different naming practices, so readers can locate them and understand where they would be used. As of now, we are left in the abstract, but visual information should never be underestimated, as it is much easier to digest, specifically in times where digitization is growing.

I suggest to invest the time to state major hypotheses on naming practices in a box, also with examples. Currently, all figures look a bit hard to the eye, one needs a lot of time to understand them. Having the journal in mind where the authors submitted their study, this is crucial for the success of the study in case of publication.

Finally, I find it unbearable that the authors do not provide any code in supplementary material or any data. Please upload your data to the Open Science Framework or to Zenodo and provide both the reviewer with access to data and code of all analyses you ran here, and second, provide us with an explicit statement regarding the data FAIRness: are the data free to share, if not, how do you guarantee the FAIRness of your study? How can we replicate your study? It should be normal by now, that also reviewers MUST have access to code during review stage, but it seems I have to demand it anyway in every second review (which is also due to editors of journals not paying attention to data FAIRness during review stage, so I do not only blame authors here, but the whole system of publication).

I suggest that the authors read a few more studies that were published in the journal they submitted their study to, in order to make acquaintance with the style and how studies are presented. My feeling is that this would greatly change the current appearance and greatly increase the chances of being accepted and also of being influential upon publication.

Version 2:

Reviewer comments:

Reviewer #1

(Remarks to the Author)

Review report for "Cross-Cultural Structures Of Personal Name Systems Reflect General Communicative Principles".

I would like to thank the authors for considering my comments and suggestions. Studying culture through personal name systems is very interesting yet challenging. I commend the authors for their efforts to address these comments and improve their work. I feel the authors have addressed most of my concerns. In the following, I only have a few remaining questions and suggestions.

1. There seems to be no easy way to figure out variations in prefix-name spelling/translation among Chinese diaspora (Woo vs Wu, and Lee vs Li). The authors noted that "our data doesn't provide the percentage of people with the prefix-name Lee being Chinese or Korean; simply including Lee as a variation of Li would distort our pattern even further." I wonder if some new calculations (at least estimations) or comparisons can be done at the country or region level. For example, in Table 4, the authors calculated the entropy for each country/region. What would the results look like if we changed Woo to Wu and Lee to Li for China? I assume that at least some results are affected simply by the translation systems, e.g., from Chinese to English (variations in spelling/translation), and this will lead to an overestimation or underestimation.

2. In the reply and page 7 of the manuscript, the authors mentioned, "Chinese languages belong to the Sino-Tibetan language family, and Korean and Japanese are both considered language isolates, which means they are not related to any other language spoken in the world today." First, I think one point should be clarified. It seems that the authors only considered written language instead of spoken language. These two are, in some cases, different. For example, some Traditional Chinese characters are adopted in Japanese with their original meaning but pronounced differently. Second, when talking about written Chinese languages or characters, it would be better to distinguish Traditional Chinese from Simplified Chinese. Now I see some mix, for example, on page 2 (Simplified) and Figure 1 (Traditional).

3. In their reply, the authors pointed out that “organizing translation conventions in the light of the information structure of names could alleviate this information loss.” What would be the simple takeaways of the main findings? Is there any practical yet straightforward way to alleviate information loss? Should we advocate writing the scientist's full name when we make references in papers? How about preserving intonations (like some French words) when translating Chinese to English, for example? I think adding more discussions about the policy implications of the findings would help.

4. By my reading, a straightforward message/takeaway is that, empirically, the given/first names are more diverse for the East Asian Cultural Sphere (EACS), while the family/surnames are more diverse for the Western countries. When referencing individuals all by their family/surnames, for example, in scientific publications, EACS people become less distinguishable from each other than Western people. This will create some inequalities, such as inequality in recognition. Moreover, translation conventions are associated with this information loss in name, where I think preserving intonations may help alleviate the information loss to some extent.

Minor issues:

5. On page 4 of the main text, the authors mentioned “the old hundred names”. I think, to be precise, it is “old hundred surnames”. It's not only about name but surname. Moreover, on the same page, the first hypothesis is very long in text. Maybe use short sentences.

6. The other half of the parentheses is missing in some sentences, for example, on page 5 “first author (where...” and on page 6 “1701-1800 (Beith, Dingwall...”.

7. Regarding Fig. 2a, I think the dip of Vietnamese-American and Korean when the name rank < 25 should be explained in detail. Regarding Fig. 2b, what does each data point represent? If each data is for one decade, annotating beside each data point would be better. If it's a binned plot, the divisions on the x-axis should be the same.

8. In Table 1, for some locales, the lower bound and the upper bound are the same. What does this tell us? Does it mean everyone in the data tends to have a different prefix-name?

9. In Figure 3, it would be better to label the top and bottom panels as (a) and (b), respectively. Also, it would be better to use warm colors to represent larger values; in other words, red color for a larger proportion in (a) and higher entropy in (b). Moreover, it seems that Table 3 can be combined with Table 4.

10. When referencing Figure 4, I think the authors should specify which panel they are referencing (as there are 4 different panels, for example, throughout the text on page 14). In Figure 4b (y-axis label), I think there should be “South Korea” and “South Korea-2”, as they are mentioned by the authors in the text and marked in Figure 4c. The same applies to “Russia-2”.

Reviewer #2

(Remarks to the Author)

I thank the authors for their attentiveness to my concerns. I am satisfied with the response.

I have one minor suggestion: it would be useful to quantify how reliable the Finnish results are regarding the subsampling to 50 births.

Einstein, like most by-names, should be high information, no?

A.4.2 has a missing Figure reference.

Reviewer #3

(Remarks to the Author)

I like the initial paper and the authors have addressed my concerns in the revised manuscript and the cover letter. I think this is an interesting and valuable contribution and I'd therefore like to see it published.

Reviewer #4

(Remarks to the Author)

I have seen an earlier version of this manuscript and I consider the new manuscript as sufficiently clear compared to the earlier one. As a result, I have no more real reservations against the publication of the study, but I would kindly ask the authors to finalize the supplementary material in the following ways:

- delete all .DS_STORE files from their GitHub repository
- make a VERSION on GitHub, so we know what state of the code went into the paper
- submit the version that went into the paper with the data to a public repository such as Zenodo (<https://zenodo.org>), so we know data has been archived for a longer term (given that GitHub is owned by Microsoft)

Version 3:

Reviewer comments:

Reviewer #1

(Remarks to the Author)

Review report for "Cross-Cultural Structures Of Personal Name Systems Reflect General Communicative Principles" by Ramscar et al.

I would like to thank the authors for their efforts in addressing my concerns, considering my suggestions, and developing their work further. In this third round of review, I feel I become an outlier among the four reviewers. Considering others' alignment and I am not a researcher who specializes in culture studies but computational social science and the science of science, I would rather not go beyond the collective consensus in this case.

In the meantime, I wanted to provide a few more notes, which do not necessarily require the authors' response because they are either very substantial or very minor. I appreciate this opportunity to learn from this study and interact with the authors who have addressed many of my initial concerns. My evaluation remains that the topic of this study is very interesting and adds to the literature of culture studies.

First, the translation systems, for example, from simplified Chinese used in mainland China to English, may affect some of the observations in this study focusing on analyzing personal names in English. As the authors argued that there was no reliable data published by official sources for them to study the size of this effect, it remains unclear how robust these observations are because the analyses are all built on the English corpus.

In their reply, the authors mentioned, "since most of the scientists in the list are from (mainland) China (foreign members are only 3% of the membership), their names are all transcribed to pinyin, and because of this, there are no spelling variations in their names: 李 will always be spelled Li, not Lee." I think the authors' claim is not supported for at least two reasons. (1) These scientists are from mainland China based on their affiliations, I guess, but they can be originally from different parts of the world or born in early times, where the translation systems can be different. For example, Professor Poo is a member of the Chinese Academy of Sciences, and the translation is "Poo" instead of "Pu". I understand the authors only sampled 425 names from the Chinese Academy of Sciences. This sample is small and may not include a particular name, but the claim is not likely true for all NAS members, for example. Second, for the general researcher population in mainland China, assuming that "李" will always be spelled "Li" is not appropriate. For example, in this publication (DOI: 10.1016/j.ins.2024.120945), it's spelled as "Lee" instead of "Li". It is fair to believe this is not common but not none.

Second, the manuscript requires substantial editing in both the text and the figures. Re the text, there are long introductions, smaller and bigger font sizes, bullet points on Pages 13, 17, and others, etc. Re figures, the font size is different across panels in one figure (e.g., Fig. 1), the font size is too small to read clearly (e.g., Fig. 4b), and all figures lack their titles, where a one-sentence title is needed for each figure in addition to the introduction for (a), (b), (c), and other panels.

Third, the implications regarding "allowing individuals more flexibility in choosing their names" (Page 16) are important for individual authors and some cultures. But at the same time, this recommendation or practice may pose substantial challenges to publishers and the development and maintenance of bibliometric databases, thinking about how difficult it will become to do name disambiguation for authors who published before and now.

Fourth, there are Pearson correlation analyses in the main manuscript, but the authors marked "n/a" in the statistics section of the reporting summary. There was very little information provided in the reporting summary. The authors should substantially improve their reporting.

Fifth, in the Data Availability Statement, I think only the first paragraph is necessary in the main manuscript. All other detailed file names (like "CA.txt, DE.txt") and variables should be in SI. The same applies to the Code Availability Statement. Moreover, the "Appendix" section should be in a standalone file as Supplementary Information.

Reviewer #2

(Remarks to the Author)

Thanks for addressing my concern. I recommend the paper for publication.

PS on the first page MIT and Tubingen are both assigned a.

We would like to thank the reviewers for the insightful and very helpful comments you made on our previous submission. We have substantially revised the paper in the light of the reviews, adding a number of new analyses and rewriting it so as to better fit the structure of the journal, with more standard methods and result sections.

After substantial revision, the version of our paper now includes:

- A significantly expanded introduction
- Detailed methods and results
- A table making our terminology clearer
- Additional analyses addressing potential confounds raised about population size and community
- Clearer graphs
- An additional analysis of Taiwanese names, to address a possible confound pointed out by R1.

We respond to the comments of each reviewer, point by point, below.

Reviewer #1 (Remarks to the Author):

Review report for “Cross-Cultural Structures Of Personal Name Systems Reflect General Communicative Principles”

1. Personal name is a product of human language, which is more often a local (original) language than widely used English. The translation of names from the original language to English may substantially affect the analysis in this study. In the example illustrated in Figure 1, the movie director Woo Yu-Sen (吳宇森), aka John Woo, was born in Guangzhou, China (at that time ROC) and grew up in Hong Kong. His English name Woo Yu-Sen is a translation from Chinese to English, which is likely following Wade-Giles romanization at that time and in Hong Kong. Nowadays, in China (mandarin in mainland China), people usually use Pinyin to do the translation. In this case, it would be Wu Yu-Sen. Here, Wu is the same as Woo (originally in Chinese), but the different translations result in different English words. I am wondering how this might affect the patterns the authors observe from the data, especially considering the translation from different languages.

Response: Thank you for pointing it out. There are indeed a wide range of variations in prefix-name spelling among Chinese diaspora. We attempted to address this by referring to a common spelling variation on Wikipedia, but we still encountered two issues. First, a variation of a prefix-name might in fact be another prefix-name, or even a prefix-name in another population. For example, Li could be spelled as Lee (e.g. Bruce Lee), which is one of the most popular Korean prefix-names. Since our data doesn't provide the percentage of people with the prefix-name Lee being Chinese or Korean, simply including Lee as a variation of Li would distort our pattern even further. Second, considering spelling variation still does not address the problem that many different prefix-names are Romanized the same (as the reviewer points out in the next point), so we find addressing this not really helpful. In the revised manuscript (Experiment 1), we included a footnote acknowledging these limitations.

To circumvent this issue, we added another dataset to Experiment 1: the 2018 census data of Taiwanese people, where prefix-names were indexed under Traditional Chinese characters. This data would give a better estimate of the entropy.

2. In some cultures, language evolves over time, and this may also affect personal names. I understand the authors used several large-scale data across different time spans, but among them, there are some recent census data. To the best of my knowledge, initially, Korean and Japanese languages are very similar to the Chinese language due to immigration from China, while there are some legislature's implementations that drive the evolution of their language, which is basically making each word very simple in its writing. This results in the challenge that, even though the pronunciation is the same, the original words in their language are very different. In other words, there is a one-to-multiple mapping. This may significantly affect the results in Figure 2, from which we can see that Korean names are concentrated on the top-ranked ones (blue line). I am wondering how language evolution may explain or affect the authors' observations.

Response: while there might be similarities among Korean, Japanese, and Chinese languages due to language contact, they are fundamentally very different. Chinese languages belong to the Sino-Tibetan language family, and Korean and Japanese are both considered language isolates, which means they are not related to any other language spoken in the world today. We have clarified this point in relation to Experiment 1.

It is true that in South Korea and Vietnam, the Chinese character-based writing systems - Hanja in Korea and Chu Nom in Vietnam - have been replaced by phonetic-based writing systems (Hangul in Korea and Chu Quoc Ngu in Vietnam), which might make prefix-names that were originally written in different characters spell the same phonetically in their current writing systems. This could lead to potential inaccuracies in the Vietnamese-American and the Chinese-American data, because both populations' prefix-names were Romanized in the US census. For the Korean data, the prefix-names were entered in Hangul (phonetically) in the original manuscript, and in the current manuscript, we fixed it by using a different dataset where prefix-names are entered in Hanja (in Chinese-based characters). The results did not change noticeably after we switched to the new dataset.

3. In Figure 2 and others, the authors used entropies to capture the distributions of different names. First, I would encourage the authors to elaborate their measures further. For example, how the entropy measure is calculated and whether using all names or just the top 50 names. Second, I am wondering if other measures would be more suitable to calculate the distributions of names, for example, the Gini index.

Response: Thank you for your suggestions. In the revised text, we added a description on how we calculated the entropy. In the original draft, we calculated the entropy of just the top 50 names, but in the revised manuscript, we calculated the entropy of the entire sample, but because of the difference in granularity in each dataset, we have also estimated an upper bound of the entropy of each sample.

We have stuck with entropy, as opposed to the Gini index, because this makes our analyses comparable and consistent with the large body of literature studying the

information-theoretic properties of human language from a communicative efficiency perspective (see Gibson, et al. 2019 for an overview).

4. The authors predicted that, during the transition period, Western prefix-name information entropy would have increased as a function of the degree to which fixed patronyms were used. In Figure 3, the authors showed the frequency at which the top three male and female prefix-names were given in England by decade 1800 — 1880 and 1900. If I understand correctly, one would expect that the entropy of name distributions would increase and the Gini index would decrease after fixed patronyms were used. Therefore, it would be better to show some results on temporal trends instead of plotting the proportion against the population. I may misunderstand this, but I would encourage the authors to elaborate more on this plot and result.

In this case, the temporal trends are tightly linked with population growth. The plot shows time (each dot is a decade), population, and entropy. We have clarified this in the new version of the paper..

5. The authors argued that “current academic publishing conventions (full hereditary-patronym, initialized bynames) lead to more ambiguity for Chinese and Korean names than for U.S. names”, and “We have shown how this potential biases against individuals with non-WEIRD names”. In my view, the evidence the authors present may not be enough to support the claim on the biases against individuals with non-WEIRD names. First, the translation of names from the original languages to English already reduces the information about names. This may have a large effect than academic publishing conventions. Second, scientists may not be a random sample from the general population, meaning that it may be hard to conclude there are biases against individuals.

We agree that scientist names may not be broadly representative but suggest that this is likely true for any domain. We picked one accessible domain (scientists/academics) and studied it. So the fact that it has distinctive properties is part of the point and not a confound in this case, as is the fact that the translation of names from the original languages to English already reduces the information about names – our point is that organizing translation conventions in the light of the information structure of names could alleviate this information loss.

Third, there are very few journals using the initials of names, primarily from the physics community (e.g., high energy physics; they usually have many authors on a signal experimental paper).

We note that the issue we point out is an issue both with initials of names (C. Darwin vs. W. Wang), as well as if it's last name only (Darwin, Wang). We now have adjusted the way we talk about citation practices in Hypothesis 3, per this comment's suggestion.

6. The authors are encouraged to structure the paper according to the journal to which they submit it. Also, the resolution of figures should be enhanced.

Thank you for the suggestions. In the revised manuscript, the texts are reformatted, and the figures are all converted to vector images.

Reviewer #2 (Remarks to the Author):

With regards to the first part of the paper, the predictions/evidence are correlational. Are there identifiable causal pathways for how fixing a by-name increases the entropy of the pre-fix name? Are there no alternative hypotheses as to why pre-fix entropy might increase over time? For example, before patronym's were fixed did people not reuse family names or names associated with a trade (Dread Pirate Roberts) until the population increased to require other names? Does the point in the paper about it not being linked to population size because Korea has a higher population than Delaware confound average population size when patronymic names were fixed? Without these considerations/discussions of alternatives the argument sounds like a just-so story.

Response: We apologize for any confusion that may have arisen on these points. When we say entropy is not linked to population size, what we mean and should have said is that an increase in population *doesn't necessarily* lead to an increase in entropy. It is when an increase in population is coupled with fixed/hereditary bynames that resulted in an increase in prefix-name entropy. We can see this by comparing the Finnish prefix-name entropy before 1800 and that after 1900. There were way more names and way more records in Finland before 1800 than after 1900, but the prefix-name entropy after 1900 is higher than that before 1800, because during this time Finland carried out regulations fixing people's bynames as hereditary. We acknowledge that in our example, comparing South Korea and Delaware is more striking but probably a little bit misleading.

The mechanism was put to test in our Experiment 2 where we looked at the Finnish birth record by time period and by geographical location. We showed that the prefix-name entropy in Finland increased with the percentage of residents taking a fixed byname, following a similar temporal and geographical pattern.

One point of contention: "An information-theoretic analysis cannot capture one critical factor in the empirical functioning of name systems: confusability." I'm not convinced this is true. It might just be imprecise writing but noisy channel coding would prefer dispersion of the form; whereas, source coding would prefer similar meanings to be compressed to similar forms for optimal meaning recovery. The charitable would argue that context further helps disambiguate. Either way confusability is an important concept to an information-theoretic analysis.

This is fair. We have changed this to "an information-theoretic analysis that treats each name atomically cannot...". Our previous analyses treated each name as discrete. The confusability relaxes this assumption and looks within words. We agree that noisy channel approaches would see some benefit to that (just as we note that there **is** some benefit to having so many "Johns" in medieval England: it allows for the re-use of an easy, common name and context disambiguates anyway).

With regards to the second part of the paper, I'm all on board with flexible names for the social reasons. We should, as a community, rethink using indexing systems that disadvantage large chunks of the world. That said, the subtext of the paper, as I read it, is that the predominant function of naming is establishing reference and the legal requirement for context-free reference might not be that important because we would still be communicatively efficient if we had contextualized naming (as in the past). This

argument doesn't address the non-reference goals that might add functional pressures on naming systems to become context-free---e.g., inheritance, ownership and provenance. I bring this up not to undercut the authors' intention but to ask for elaboration on what kind of flexibility they are asking for and what the implications would be beyond reference.

We subsume these goals (which we agree have some advantages in administration, ownership, etc) in the discussion of the goals of the government/bureaucratic systems that impose legalized naming systems. We agree that these alternative goals are at least partially an explanation for the patterns we observe and it's fair to say that there is a tradeoff in optimization for those goals vs. optimization for communication in context. But we do not formalize that tradeoff here and leave it for future work.

Minor notes:

Pg 3, The predictions for the third hypothesis are vague compared to those actually tested.

The hypothesis has now been made clearer.

The main text before Figure 2 does not introduce the Northern England dataset.

We have substantially expanded the methods sections now, which more clearly spells out the methods and data sets.

In Figure 2, it would be helpful to have different markers for EACS and Western cultures.

Figure 2 is now substantially changed.

At first mention, it's unclear what's meant by indexing and why it's important. The examples clear it up though.

Pg 7 not initialized → initialized

Fixed

The K2 and R2 conditions in Figure 5 need to be explained better.

We now address this.

Reviewer #3 (Remarks to the Author):

I really liked the paper and would like to see it published. I did have one methodological concern, and then a couple of points of confusion that could be cleared up with a bit of an edit.

Response: thank you for your comments.

The substantive concern: regarding the low entropy of pre-modern Western prefix names, I can sort of buy that population size not the only factor here, since you take the 50 most frequent names in each data set and also, as you point out, population size doesn't correlate perfectly with entropy. But I did wonder about other features that might artificially suppress the prefix name entropy of the smaller data sets. For instance, what about relatedness/non-independence in the Scottish samples - to what extent are these all date from a small set of families in those towns/villages? It's old fashioned these days (like 1-2 generations out of date), but it's not uncommon in the UK to have children take the same first name as a parent or parent's sibling - my dad is named after his dad, the same is true of several of my friends, I think this used to be very standard and quite strictly applied, with middle names used to individuate (and people in practice being referred to by their middle name). Even if people are not following this old tradition, they may copy names they see around and like - I have the same name as my uncle, my daughter has the same name as my sister-in-law, apparently for this reason. Furthermore I think the tradition of naming sons after fathers, or copying names from elsewhere in the family, is itself a tradition that varies by families. You also get weird naming clusters (e.g. weirdly loads of kids named "Kai" in my son's class) which I think must reflect independent name copying from a locally prominent exemplar. Those sort of effects should suppress name variation, and could (I think) have larger effects in smaller populations, e.g. where everyone is copying an already-small pool of names or naming practices. Can you discount those sorts of effects as leading to misleading parallels in prefix name entropy, e.g. in Figure 2?

Yes, we do think these effects are playing a name here *within* communities. Clustering of names is likely an explanation for overlap. But we also note that our British towns (e.g., Dingwall, Govan, Beith) are quite far apart by 1700s standards (see below) and would have been outside the distances people normally traveled. So we think it's unlikely there are clustering effects *across* towns.

Nonetheless, it might be interesting in future work to build models that explicitly account for the non-i.i.d. nature of naming practices within a community.

[Figure Redacted]

Points of my confusion where a little clarification would help:

Where you say “Information theory shows how combining codewords of different frequencies (John, Claude) with other codewords conditioned on them (Smith, Shannon) can allow probabilistic codes that near optimally balance the competing demands of encoding individuals and communicating identities to be defined” - I think it would be nice to illustrate this somehow since an intuitive understanding of why this is the case would be helpful. In figure 1, I also wasn't sure if the lower-information word should come first by design, or if the increase in information in later words is because of conditioning/combinatorics.

We've clarified Figure 1.

Regarding the increase in prefix name entropy in Finland - wouldn't you also predict a decrease in entropy in bynames in the same period? I think that's what figure 4 upper is intended to show, but it's slightly indirect in that it shows instead use of father's hereditary patronymic. Can you show the decrease in byname entropy directly, or is there a reason why this can't be done? I wondered if part of the problem was that people were simply not using bynames in the pre-modern data - but in that case, how was the individuating function of names achieved?

That's right: we don't have access to bynames until the hereditary bynames are introduced. In some cases, we have the father's prefix-name, which is largely how the disambiguating may have been done (e.g., John, son of Daniel). But it's not directly comparable and so we use proportion of byname in the records as a proxy for fixed hereditary patronyms.

I think the section on the problem with current conventions around initialising given names is really interesting - but I got quite confused here, because you bring in the terminology "Family" and "Given" names, and then (I think) what you are calling the prefix name for EACS authors is used second in scientific citations - e.g. Woo Yu-Sen becomes Y. Woo rather than W. Yu-Sen (which would actually be preferable for individuation purposes, if I am reading your results correctly). There are places where you also seem to mix the two different terminologies, i.e. describing the convention as "full hereditary- patronym, initialized bynames" - isn't it "full hereditary patronym, initialised given name" or "full hereditary patronym, initialised byname for westerners, initialised prefix name for EACS individuals"? I also wasn't sure what I was supposed to take from the right-hand panel of figure 5. Anyway, my uncertainty around the terminology and what I was to take from Figure 5 made me doubt that I actually had got your conclusion correctly - I *think* you are saying that e.g. while C. Darwin works nicely for WEIRD names, something like W. Yu-Sen would be more individuating for EACS names (and Y. Wu is very bad for them).

We have clarified this, as discussed above.

Two minor additional quibbles on figures: in figure 2, it would be better to use less code-internal variable names in the legend, and definitely not use CA and DE abbreviations, which will be much less familiar as US states to non-US readers; in figure 3, please don't use scientific notation in the x-axis - just say "Population (millions)" and give easier-to-read numbers.

Thank you for the suggestions. In Figure 2, the states are represented by their full names. In Figure 3, the population is now presented in millions.

That's it. As I said, if I am following it correctly, I think the paper is really interesting and I'd like to see it published if it can be revised to 1) spare the reader some of the confusions I had and 2) deal (in supplementary materials I expect) with my concern re. the effects of name copying in small populations.

Thank you for your comments.

Reviewer #4 (Remarks to the Author):

I think this study has an enormous potential, given that it discusses a topic that is rarely analyzed with quantitative approaches, but which the authors show to have an impact with respect to the naming practice in publications in a globalizing world.

In its current form, however, the study loses a lot of its potential due to the way the results and the study are presented. It is very difficult to follow the reasoning and to understand what studies were undertaken, since the authors do not structure their study in a traditional way that would also be common for readers of the journal. This means, the authors should -- if they follow my suggestion -- completely re-do their main structure of the article, by separating materials and methods from analyses, and stating hypotheses clearly in the introduction and then discussing their findings in the conclusion. In the current form, the paper may be interesting for journals with another peer group, but the journal where the authors submitted typically looks at particular findings done by applying research methods, which need to be presented in such a structured way.

Agreed. We've revised the structure of the overall paper to fit the more traditional model of methods + results.

Furthermore, I miss examples in figures or boxes or tables of how name practices differ. I think it would be easy to come up with some nice illustrations, maybe even taking world maps where one plots different naming practices, so readers can locate them and understand where they would be used. As of now, we are left in the abstract, but visual information should never be underestimated, as it is much easier to digest, specifically in times where digitization is growing.

I suggest to invest the time to state major hypotheses on naming practices in a box, also with examples. Currently, all figures look a bit hard to the eye, one needs a lot of time to understand them. Having the journal in mind where the authors submitted their study, this is crucial for the success of the study in case of publication.

We've revised the text and figures to be more in line with the journal. We've added a clearer box explaining our naming terminology.

Finally, I find it unbearable that the authors do not provide any code in supplementary material or any data. Please upload your data to the Open Science Framework or to Zenodo and provide both the reviewer with access to data and code of all analyses you ran here, and second, provide us with an explicit statement regarding the data FAIRness: are the data free to share, if not, how do you guarantee the FAIRness of your study? How can we replicate your study? It should be normal by now, that also reviewers MUST have access to code during review stage, but it seems I have to demand it anyway in every second review (which is also due to editors of journals not paying attention to data FAIRness during review stage, so I do not only blame authors here, but the whole system of publication).

Our code and analyses were intended to be available in a github (linked in the original draft), but we now notice we did not make it public. This was simply an oversight!

Response to reviewers

We thank the reviewers again for their comments, and we appreciate the chance to submit another revision. Below is our response to each of points raised.

REVIEWER COMMENTS

Reviewer #1 (Remarks to the Author):

Review report for “Cross-Cultural Structures Of Personal Name Systems Reflect General Communicative Principles”.

I would like to thank the authors for considering my comments and suggestions. Studying culture through personal name systems is very interesting yet challenging. I commend the authors for their efforts to address these comments and improve their work. I feel the authors have addressed most of my concerns. In the following, I only have a few remaining questions and suggestions.

Thanks for your helpful comments!

1. There seems to be no easy way to figure out variations in prefix-name spelling/translation among Chinese diaspora (Woo vs Wu, and Lee vs Li). The authors noted that “our data doesn’t provide the percentage of people with the prefix-name Lee being Chinese or Korean; simply including Lee as a variation of Li would distort our pattern even further.” I wonder if some new calculations (at least estimations) or comparisons can be done at the country or region level. For example, in Table 4, the authors calculated the entropy for each country/region. What would the results look like if we changed Woo to Wu and Lee to Li for China? I assume that at least some results are affected simply by the translation systems, e.g., from Chinese to English (variations in spelling/translation), and this will lead to an overestimation or underestimation.

This is a great suggestion but unfortunately we don’t have the data needed to carry out this analysis, as there was no reliable data published by official sources. We see two issues with testing the effect of spelling variations on Chinese scientist names. First, since most of the scientists in the list are from (mainland) China (foreign members are only 3% of the membership), their names are all transcribed to pinyin, and because of this, there are no spelling variations in their names: 李 will always be spelled Li, not Lee. Second, the problem with the Chinese to English transliteration system is not just that one name can be spelled multiple ways, but also that multiple names can be transliterated into one. Without a good database that is publicly / legally accessible, it will be very hard to conduct this analysis. Due to privacy concerns, most of the census data is anonymized, which, by definition, has people’s

names removed. As the editor will recall, prior to going into review, we had gone through an exhaustive procedure to ensure all the included data was legitimately accessed and used.

2. In the reply and page 7 of the manuscript, the authors mentioned, “Chinese languages belong to the Sino-Tibetan language family, and Korean and Japanese are both considered language isolates, which means they are not related to any other language spoken in the world today.” First, I think one point should be clarified. It seems that the authors only considered written language instead of spoken language. These two are, in some cases, different. For example, some Traditional Chinese characters are adopted in Japanese with their original meaning but pronounced differently. Second, when talking about written Chinese languages or characters, it would be better to distinguish Traditional Chinese from Simplified Chinese. Now I see some mix, for example, on page 2 (Simplified) and Figure 1 (Traditional).

Thank you for the clarifications. With regards the first point, we note first that we do not discuss Japanese in the paper, and second, that although Korean and Vietnamese have indeed borrowed a lot of Chinese words and to some extent retained their pronunciation over the years, their basic vocabulary is different, which means that these languages are unlikely to share a common ancestor. Similarly, English borrowed a large amount of words from Old Norse, French, Greek, and Latin in different periods of times, but this does not make English a Romance or Hellenic language.

With regards the second, we agree this was potentially confusing: we have fixed the mixing of two different Chinese scripts (陈宏 -> 陳宏).

3. In their reply, the authors pointed out that “organizing translation conventions in the light of the information structure of names could alleviate this information loss.” What would be the simple takeaways of the main findings? Is there any practical yet straightforward way to alleviate information loss? Should we advocate writing the scientist’s full name when we make references in papers? How about preserving intonations (like some French words) when translating Chinese to English, for example? I think adding more discussions about the policy implications of the findings would help.

Thank you for your suggestions. We have fleshed out the policy implications, while being mindful of the editor’s request not to overstep the empirical nature of our claims. We can’t make claims about which policies would be best since we don’t test policies explicitly, although we do think our empirical work here points to further studies that could be done in this direction.

4. By my reading, a straightforward message/takeaway is that, empirically, the given/first names are more diverse for the East Asian Cultural Sphere (EACS), while the family/surnames are more diverse for the Western countries. When referencing individuals all by their

family/surnames, for example, in scientific publications, EACS people become less distinguishable from each other than Western people. This will create some inequalities, such as inequality in recognition. Moreover, translation conventions are associated with this information loss in name, where I think preserving intonations may help alleviate the information loss to some extent.

We agree! And we hope that our paper will spur positive policy changes and more work on this from not just the linguistic perspective, but the social science perspective. We mention these implications more now, but adhere to the editor's request that we not overstep.

Minor issues:

5. On page 4 of the main text, the authors mentioned "the old hundred names". I think, to be precise, it is "old hundred surnames". It's not only about name but surname.

We think the word 'surnames' is misleading, as it means 'the name that comes after'. To avoid confusion, we adhere to our byname/prefix name terminology and want to avoid using "surnames".

Moreover, on the same page, the first hypothesis is very long in text. Maybe use short sentences.

6. The other half of the parentheses is missing in some sentences, for example, on page 5 "first author (where..." and on page 6 "1701-1800 (Beith, Dingwall...".

Thanks very much for pointing this out! We have fixed it.

7. Regarding Fig. 2a, I think the dip of Vietnamese-American and Korean when the name rank < 25 should be explained in detail. Regarding Fig. 2b, what does each data point represent? If each data is for one decade, annotating beside each data point would be better. If it's a binned plot, the divisions on the x-axis should be the same.

Each data point represents the year in which the population was surveyed. We now have included such information. We don't have a very clear explanation of the dip for rank <25, but think this would be an interesting avenue for further research.

8. In Table 1, for some locales, the lower bound and the upper bound are the same. What does this tell us? Does it mean everyone in the data tends to have a different prefix-name?

This simply tells us these datasets are exhaustive: the name of every person surveyed was present in these dataset. The entropy calculated from these datasets is the entropy of the samples, and the lower bound is the same as the upper bound. In contrast, in some other datasets, possibly due to privacy concerns, only names that are shared by more than 5 people

were present in the dataset, and the entropy calculated from these datasets is merely a lower estimate, so the lower bounds and the upper bounds are different.

This does not mean everyone from exhaustive datasets tends to have a different name, since we don't have access to the rest of the non-exhaustive datasets.

9. In Figure 3, it would be better to label the top and bottom panels as (a) and (b), respectively. Also, it would be better to use warm colors to represent larger values; in other words, red color for a larger proportion in (a) and higher entropy in (b). Moreover, it seems that Table 3 can be combined with Table 4.

We have implemented your suggestions regarding Figure 3. We prefer to keep Tables 3 and 4 separate since they are showing different things.

10. When referencing Figure 4, I think the authors should specify which panel they are referencing (as there are 4 different panels, for example, throughout the text on page 14). In Figure 4b (y-axis label), I think there should be "South Korea" and "South Korea-2", as they are mentioned by the authors in the text and marked in Figure 4c. The same applies to "Russia-2".

Thank you for your suggestions. We have now included Korea-2 and Russia-2 in Figure 4b, Figure 4c, and Figure 4d.

Reviewer #2 (Remarks to the Author):

I thank the authors for their attentiveness to my concerns. I am satisfied with the response.

I have one minor suggestion: it would be useful to quantify how reliable the Finnish results are regarding the subsampling to 50 births.

Thank you for your suggestions. We did two additional analyses. In the first one, we repeated our analyses regarding Finnish prefix-name entropy 500 times. In the second one, we repeated our analyses 500 times, but in each we subsampled 100 births instead of 50 births when calculating the prefix-name entropy. All the results remained unchanged. In the revised manuscript, we described the methods and the results in Appendix A.7.

Einstein, like most by-names, should be high information, no?

Thank you for pointing this out. We have fixed it.

A.4.2 has a missing Figure reference.

Thank you. We have now fixed it.

Reviewer #3 (Remarks to the Author):

I like the initial paper and the authors have addressed my concerns in the revised manuscript and the cover letter. I think this is an interesting and valuable contribution and I'd therefore like to see it published.

Thank you for the comments.

Reviewer #4 (Remarks to the Author):

I have seen an earlier version of this manuscript and I consider the new manuscript as sufficiently clear compared to the earlier one. As a result, I have no more real reservations against the publication of the study, but I would kindly ask the authors to finalize the supplementary material in the following ways:

- delete all .DS_STORE files from their GitHub repository
- make a VERSION on GitHub, so we know what state of the code went into the paper
- submit the version that went into the paper with the data to a public repository such as Zenodo (<https://zenodo.org>), so we know data has been archived for a longer term (given that GitHub is owned by Microsoft)

Thank you for the suggestions. We made a version on Github (v1.1) and the relevant materials have been archived at Zenodo (DOI: <https://zenodo.org/doi/10.5281/zenodo.13755110>)

Response to Reviewers

We thank the reviewers and the editor again for their comments, and we appreciate the chance to submit another revision. Below is our response to each of points raised.

Response to the editor's comments

- Please provide a point by point response to the remaining reviewer comments. In response to Reviewer 1, please include in the manuscript a discussion of limitations related to the dependence on an English Corpus, inability to test effects of variations in Chinese spelling, and the small sample
 -
 - We take "English Corpus" here to mean the transliteration from Chinese characters to a Latin alphabet.
 -
 - As we have sought to further clarify in the manuscript, a large part of the motivation for the study we report in Experiment 3 is to examine exactly the information loss that results from this kind of transliteration – i.e., the information lost when Chinese names are rendered into pinyin (i.e., Latinized) and communicated in scientific publishing: "To test the hypotheses, we obtained cross-cultural samples of scientific communities by scraping membership lists of academic/scientific societies in the United States, China, Korea, France, Finland, and Russia. We explicitly chose these samples in order to examine the real-world impact of current indexing conventions (and hence the transliteration of character-based Chinese and Korean names to the Latin alphabet) in a domain where the communication of identities is critical, namely academic publishing."
- Note that to facilitate direct comparisons, we randomly downsampled from the populations of the larger societies so as to have evenly matched sample sizes in all our groups – this still resulted in a situation where even the downsampled datasets represented over 15% of their respective populations. Given the large effects across populations, we do not think 15% of the total population of relevant individuals is too small to draw meaningful conclusions. It is also important to consider that what drives these results is that in, say Chinese, a small set of prefix-names cover a large amount of the sample as compared to the US; given the shape of the name distribution, this will always be the case in any reasonably sized subsample, regardless of how that sample is drawn.

- We think this comment thus also addresses the reviewer's concern about the sample size, which is appropriate for testing effects across groups (although we agree it won't necessarily include all variations of a given name). We no longer include the claim **“since most of the scientists in the list are from (mainland) China (foreign members are only 3% of the membership), their names are all transcribed to pinyin, and because of this, there are no spelling variations in their names: 李 will always be spelled Li, not Lee.”**
-
- However, as we discuss in response to the reviewer's comments, we are inclined to agree that our method for extracting a Chinese American sample from Census data In Experiment 1 was problematic, especially because we did not account for the spelling variations of the same last name (e.g., the prefix-name 陈 (simplified) / 陳 (traditional) can be spelled as ``Chen" or ``Chan" in our counts. Given that our Taiwanese dataset contains counts for prefix-names that are recorded using traditional Chinese characters, we believe that it offers a far better insight into modern Chinese name distributions, and in the light of the reviewer's comments, we have deleted reference to the problematic Chinese American dataset from the article, and instead focus our analysis on the untranslated Taiwanese dataset.
-

Response to the Reviewer 1's comments

- I would like to thank the authors for their efforts in addressing my concerns, considering my suggestions, and developing their work further. In this third round of review, I feel I become an outlier among the four reviewers. Considering others' alignment and I am not a researcher who specializes in culture studies but computational social science and the science of science, I would rather not go beyond the collective consensus in this case.
-
- **Response: thank you for your review.**
- In the meantime, I wanted to provide a few more notes, which do not necessarily require the authors' response because they are either very substantial or very minor. I appreciate this opportunity to learn from this study and interact with the authors who have addressed many of my initial concerns. My evaluation remains that the topic of this study is very interesting and adds to the literature of culture studies.
-
- **Response: thank you**

- First, the translation systems, for example, from simplified Chinese used in mainland China to English, may affect some of the observations in this study focusing on analyzing personal names in English. As the authors argued that there was no reliable data published by official sources for them to study the size of this effect, it remains unclear how robust these observations are because the analyses are all built on the English corpus.
-
- By “English Corpus”, we again take this to mean the transliteration from Chinese characters to a Latin alphabet.
-
- With regards to Experiment 1, we note that although the Chinese American sample extracted that we from US Census data was Latinized (and in fact, rendered in the English alphabet due to US census data formatting rules), the Taiwanese dataset we analyzed is not a translation. Rather it contains counts for prefix-names that are recorded using traditional Chinese characters.
-
- We agree that our method for extracting a translated Chinese American sample from Census data is problematic, especially because we did not account for the spelling variations of the same last name (e.g., the prefix-name 陈 (simplified) / 陳 (traditional) can be spelled as “Chen” or “Chan” in our counts. In the light of the reviewer’s justified concerns about evaluating name distributions using translated data, and since our Taiwanese dataset offers us an insight into modern Chinese name distributions that avoids the limitations associated with our problematic Chinese American dataset, we decided that the most straightforward way of addressing the reviewer’s concerns was to delete any reference to the latter from the article, and to focus our analysis of the more reliable, untranslated data in the former.
-
- By contrast, as we noted above, and as we have sought to further clarify in the manuscript, a large part of the point of the study reported in Experiment 3 is to examine the information loss that results from this kind of transliteration, and thus we do not think that the dataset we use in this experiment is limited in this way.
-
- In their reply, the authors mentioned, **“since most of the scientists in the list are from (mainland) China (foreign members are only 3% of the membership), their names are all transcribed to pinyin, and because of this,**

there are no spelling variations in their names: 李 will always be spelled Li, not Lee." I think the authors' claim is not supported for at least two reasons. (1) These scientists are from mainland China based on their affiliations, I guess, but they can be originally from different parts of the world or born in early times, where the translation systems can be different. For example, Professor Poo is a member of the Chinese Academy of Sciences, and the translation is "Poo" instead of "Pu". I understand the authors only sampled 425 names from the Chinese Academy of Sciences. This sample is small and may not include a particular name, but the claim is not likely true for all NAS members, for example. Second, for the general researcher population in mainland China, assuming that "李" will always be spelled "Li" is not appropriate. For example, in this publication (DOI: 10.1016/j.ins.2024.120945), it's spelled as "Lee" instead of "Li". It is fair to believe this is not common but no none.

- - We agree there might be some amount of variance in the transliteration, but we do not take this to be a critical issue with our analysis. In fact, in a sense, this is the object of our study: the information content of the transliterated names (no matter how they are generated). We have clarified this point on page 13.
 - Also, as we noted above, given the large effects across populations, we do not think 15% of the total population of relevant individuals is too small to draw meaningful conclusions. Note that what drives these results is that in Chinese, a small set of prefix-names cover a large amount of the sample; given the shape of the name distribution, this will always be the case in any reasonably sized subsample, regardless of how that sample is drawn.
- Second, the manuscript requires substantial editing in both the text and the figures. Re the text, there are long introductions, smaller and bigger font sizes, bullet points on Pages 13, 17, and others, etc. Re figures, the font size is different across panels in one figure (e.g., Fig. 1), the font size is too small to read clearly (e.g., Fig. 4b), and all figures lack their titles, where a one-sentence title is needed for each figure in addition to the introduction for (a), (b), (c), and other panels.
- - Response: we apologize for the stylistic variations. We fixed the issue about variations of font sizes in figure captions. We increased the text size in Figure 4. We also added titles to each figure.

- Third, the implications regarding “allowing individuals more flexibility in choosing their names” (Page 16) are important for individual authors and some cultures. But at the same time, this recommendation or practice may pose substantial challenges to publishers and the development and maintenance of bibliometric databases, thinking about how difficult it will become to do name disambiguation for authors who published before and now.
 - This is a fair point and we now acknowledge that, while there are clear benefits to what we propose, there are also limitations. We do not aim to make a policy proposal but rather to give empirical evidence of an issue that could be of relevance to stakeholders in scientific publishing. We have rewritten the passage that the reviewer highlighted to reflect this more nuanced perspective.
- Fourth, there are Pearson correlation analyses in the main manuscript, but the authors marked “n/a” in the statistics section of the reporting summary. There was very little information provided in the reporting summary. The authors should substantially improve their reporting.
 - We have checked “yes” in the revised reporting summary. The Pearson correlations in the paper are calculated using standard statistical libraries in R.
- Fifth, in the Data Availability Statement, I think only the first paragraph is necessary in the main manuscript. All other detailed file names (like “CA.txt, DE.txt”) and variables should be in SI. The same applies to the Code Availability Statement. Moreover, the “Appendix” section should be in a standalone file as Supplementary Information.
 - Response: thank you for pointing it out. We now move all the details to a README file on Github from the Data Availability Statement.